

# Wind inflow observation from load harmonics

Marta Bertelè[1], Carlo L. Bottasso[1,2], Stefano Cacciola[2], Fabiano Daher Adegas[3], and Sara Delport[3]

[1]Wind Energy Institute, Technische Universität München, Garching bei München, D-85748 Germany
[2]Dipartimento di Scienze e Tecnologie Aerospaziali, Politecnico di Milano, Milano, I-20156 Italy
[3]GE Global Research, Garching bei München, D-85748 Germany

*Correspondence to:* C.L. Bottasso (carlo.bottasso@tum.de)

**Abstract.** The wind field leaves its fingerprint on the rotor response. This fact can be exploited to use the rotor as a sensor: by looking at the rotor response, in the present case in terms of blade loads, one may infer the wind characteristics. This paper describes a wind state observer that estimates four wind parameters, namely the vertical and horizontal shears and the yaw and upflow misalignment angles, from out-of-plane and in-plane blade bending moments. The resulting observer provides on-rotor wind inflow characteristics that can be exploited for wind turbine and wind farm control. The proposed formulation is evaluated by extensive numerical simulations in turbulent and non-turbulent wind conditions using a high-fidelity aeroservoelastic model of a multi-MW wind turbine.

## 1   Introduction

The wind blowing over a wind turbine rotor leaves its own specific fingerprint on the machine response. If this information is rich enough and if the wind turbine response can be measured (for example in terms of loads), then one may think of turning the rotor into a wind sensor and use it to infer the wind inflow.

Measurements of the rotor inflow during operation are attractive for a number of reasons, as they may find a wide range of applications. For example, information on the wind speed at the rotor disk is typically useful for wind turbine control, as controller behavior is often scheduled as a function of wind speed. In addition, knowledge of the wind direction with respect to the rotor is necessary not only to maximize energy harvesting, but also because operating with excessive misalignment increases loading. Wake redirection strategies (Fleming et al., 2014; Jimenez et al., 2010) deliberately point the rotor away from the wind, with the goal of deflecting the wake and reducing its interaction with downstream machines, a control strategy that again requires good knowledge of the wind direction in order to be implemented. Upflow can change significantly in complex terrain applications and, if known, it can be used to reduce loading. The presence of an impinging wake, shed from an upstream wind turbine, may result in high horizontally sheared flow at the rotor disk. Turbulence intensity (TI) and vertical shear may give indications on the characteristics of the atmosphere, information that can be used for optimizing wind turbine and wind farm control behavior. More in general, by turning each wind turbine in a wind sensor capable of measuring the local inflow characteristics, one may build a more complete picture of the wind flow within a power plant, providing information that may possibly have a variety of uses.



Unfortunately, high quality information of the wind inflow is in general difficult to obtain. On board wind turbines, wind speed is typically measured by cup or sonic anemometers, while direction is provided by wind vanes. These sensors invariably suffer from a number of disturbances, such as the presence of the nacelle, blade passing and wake-induced flow deformation. Although most of these effects can be mitigated by the use of calibrated transfer functions, filtering and ad hoc processing of

the raw measurements, all these sensors provide only local information at the specific point in the flow where they are installed. For control applications, it is clear that rotor-equivalent information is in general more appropriate than local data, as in fact what determines the overall rotor response is what is felt by the whole rotor rather than what takes place at a specific point. Additionally, certain wind characteristics can only be defined over a rotor disk and do not have point-wise equivalents, as for example shears. Met-masts, being equipped with multiple wind sensors away from the rotor, do not suffer from some of these

issues. However, the problem of mapping the information from a met-mast to the rotor disk of a wind turbine is in general very difficult to solve, and it will clearly be always prone to possibly severe inaccuracies. With LiDARs (light detection and ranging), laser-based sensing technology is rapidly becoming a game changer, and other remote sensing solutions are also very promising. While their potential is clearly very real and will probably have a deep impact on wind energy technology, these devices are still not in widespread use, mostly because of cost, reliability, availability and life-time issues.

In this scenario, wind sensing by using the rotor response seems to offer an attractive alternative. In fact, any wind property estimated from the rotor response will be non-local and rotor-effective, in contrast to local sensors. In addition, this approach provides measurements directly at the rotor disk, this way avoiding the need for mapping flow characteristics from one point to another.

The rotor-effective wind speed estimator (Van der Hooft and Engelen, 2004; Soltani et al., 2013) is one of the first examples

of the use of the rotor response for estimating wind characteristics. In that case, the idea is to use the dynamic torque balance equation: based on a map of the aerodynamic torque (or power) of the rotor over the operating envelope of the machine, one may solve this equation in terms of the unknown wind speed, assuming that the other operational parameters (rotor speed, pitch setting, electrical torque) are measured at each instant of time.

This idea was first generalized by Bottasso et al. (2010), which introduced the concept of *the rotor as an anemometer*. Instead

of using the single torque balance, additional equations of dynamic equilibrium of the machine were used, including the tower and blade degrees of freedom. As multiple equations are now available, multiple wind states can be estimated in addition to wind speed. Although attractive, the need to estimate some wind turbine states resulted in a fairly complicated formulation.

A much simpler approach was developed later on in Bottasso and Riboldi (2014). In that case the idea was not to use the equations of dynamic equilibrium, but rather to consider the steady state response of the machine. Specifically, the approach

exploited the fact that steady wind conditions are associated with a periodic response of the wind turbine. Therefore, a load-wind model was derived linking the $1{\times}$Rev harmonics of the blade out and in-plane bending moments with the wind vertical shear and yaw misalignment. A simple blade flapping model was used to derive and justify the structure of the model, while, for accuracy, its actual coefficients were obtained by identification from a higher fidelity aeroservoelastic model of the wind turbine or directly from field tests. A validation of the observer using field data was described in Bottasso and Riboldi (2015) using the

Control Advanced Research Turbine (CART3) (Fleming et al., 2011; Bossanyi et al., 2009). Results indicated a significantly



higher correlation of the observer results with respect to a met-mast, assumed as ground truth, than for the on-board nacelle anemometer and wind vane. Notwithstanding these very promising results, the same study also showed a marked sensitivity of the results on the wind upflow angle, indicating the probable need for a richer description of the wind field.

Following the idea described in Bottasso et al. (2010), an estimator based on a linearized wind turbine model was proposed in Simley and Pao (2014). The formulation used generator speed, fore-aft nacelle acceleration and collective, cosine and sine components of the blade out-of-plane bending moments to estimate, by a Kalman filter, the equivalent wind speed together with the linear vertical and horizontal shears. That study demonstrated the performance of the formulation using non-turbulent wind fields that were exactly parameterized by the assumed wind states. However, the effects of unmodeled wind characteristics (as for example turbulence and yaw or upflow misalignments) were not considered.

The concept of the wind turbine as a wind sensor was recently extended to the detection of wake impingement in Bottasso et al. (2015, 2016), where loads are used to detect if and where a wake shed by an upstream wind turbine interferes with the rotor.

Motivated by the very promising validation results in the field, the present paper extends and improves the formulation of Bottasso and Riboldi (2014), with the goal of addressing some of its weaknesses.

First, extensive numerical experiments have shown that the load-wind model on which the estimator is based must consider at least four wind states instead of two, i.e. the two yaw misalignment and upflow angles, as well as the two horizontal and vertical shears. These four states, together with the mean rotor-equivalent speed, represent the lowest order full approximation of the wind inflow at the rotor disk: the two angles give the orientation of the mean wind vector with the rotor axis, while mean speed and two shears describe a tilted planar (or mixed linear-exponential, depending on the type of shears considered) inflow. All of these states leave significant signatures in the low frequency response of the rotor. Therefore, failure to include one of them in the model will invariably create inaccuracies in the others.

Second, the paper shows that estimators of these four states should be limited to the use of the $1\times$Rev response. In fact, although $2\times$Rev harmonics are indeed excited by the four states, these same harmonics are also very significantly excited by turbulence, i.e. by higher order wind states (describing a non-planar inflow distribution over the rotor disk). As it is not possible to distinguish the part of the $2\times$Rev response caused by the four wind states from the part caused by turbulence, inclusion of this higher order response will result in significant pollution of the estimates.

Third, the paper compares both a linear and a nonlinear (quadratic) load-wind model. Both models are scheduled with respect to wind speed, in order to account for the different characteristics of a wind turbine in its wind speed operating range. Numerical experiments show that the two are very similar in performance, with a small improvement in accuracy for the nonlinear model over the linear one.

Fourth, experience has shown that angles (yaw misalignment and upflow) are significantly more difficult to estimate than shears. The paper explains the reason for this behavior from two different perspectives. From a mathematical point of view, an a priori analysis based on the Singular Value Decomposition (SVD) demonstrates that angles have a lower level of observability than shears, implying that any small error or perturbation (in the model, in the measurements, in the numerical solution, etc.) will be significantly amplified during the model inversion necessary for the estimation of the wind states. From a physical point





of view, this is also easily explained in terms of sensitivity of angle of attack changes at the blade section to wind state changes. As angles of attack (and hence loads) change less in response to angle changes than to shear changes, then angles are harder to estimate than shears when looking at rotor loads.

Finally, the paper demonstrates the performance of the estimator by extensive numerical simulations performed with a high fidelity aeroservoelastic model of a multi-MW wind turbine. The numerical results illustrate the excellent ability of the proposed formulation to follow rapid fluctuations of shears. The same results also show a very interesting behavior of the angle estimators. In fact, although angle estimates are indeed in general polluted by oscillations that depend on turbulence level, their mean errors are significantly low. An analysis that considers the probability distributions of wind speed and turbulence intensity at a given site, shows that the expected average inaccuracy of the angle estimates is remarkably low, i.e. less than one degree. This means that angles, although apparently oscillatory on short time horizons, can be followed quite precisely in their mean value changes.

The paper is organized according to the following plan. Section 2 presents the formulation of the observer, first introducing load-wind models that relate wind states and blade harmonics, then describing the identification of the model parameters by a system identification approach, and finally inverting the model to give wind states when loads are measured. A first set of simulations is used to motivate the limitation of the load vector to the 1×Rev harmonics. To this end, the simulation environment is briefly introduced together with the aeroservoelastic mathematical model of a wind turbine, used throughout the entire work to support all numerical experiments. Section 3 is devoted to an a priori observability analysis of the wind parameters using the SVD, followed by a concise summary of the expected observer behavior given in §3.2. Extensive testing of the proposed method in non-turbulent and turbulent wind conditions is given in Section 4. Finally, Section 5 completes the manuscript, listing the main conclusions and giving possible further improvements to the methodology.

## 2 Formulation

### 2.1 Wind anisotropy and its parameterization

The development of the proposed wind inflow observer is inspired by the idea of using the wind turbine as an anemometer. In this sense, wind is not only the source of energy to be harvested but also one of the principal factors affecting the wind turbine response. Specifically, the present observer is based on the lowest load harmonics. Although other response indicators could be used in principle, as for example accelerations, loads are considered in this work because they are now often measured on board modern large wind turbines for enabling load-feedback control, and load sensors will probably be standard equipment available on most future machines.

In order to understand the connection between blade loads and wind characteristics, consider now two different constant-in-time wind fields. A first wind field is axially symmetric with respect to the rotation axis of the wind turbine rotor, while the second is not, both in magnitude and/or direction. In the second –anisotropic– case, differences in speed and/or direction over the rotor disk may be due to wind shears (both vertical and horizontal) and/or misalignments with the wind direction (both due to yawed flow and upflow caused by rotor uplift, terrain orography, etc.). In the axially symmetric case, the angle of



attack experienced by the blade sections during their azimuthal travel over the rotor disk will be constant; hence, the resulting aerodynamic loads will also be constant. In the non axially-symmetric case, any anisotropy in the wind will cause periodic fluctuations in the angle of attack at the blade sections, and hence periodic loads. Amplitude and phase of such loads will depend on the wind field at the rotor disk, and on the aeroelastic characteristics of the rotor blades. Therefore, amplitude and

phase of the periodic loads carry information on the wind anisotropy at the rotor disk. This fact can be readily verified with simplified mathematical models of a rotating blade in an anisotropic wind field, as for example the classical flapping and lagging blade model developed in Eggleston and Stoddard (1987). Using such a model, Bottasso and Riboldi (2014) suggested a linear structure for a blade response-based observer of cross-flow and vertical shear.

In this work, the wind field anisotropy is parameterized using four variables (termed *wind states* in the following): the vertical

shear exponent $\kappa_v$ and horizontal linear shear $\kappa_h$, and the two angles $\phi$ and $\chi$, measuring respectively the yaw misalignment and upflow. These quantities are collected in the wind state vector $\boldsymbol{\theta}$, defined as

$$\boldsymbol{\theta} = (\phi, \kappa_v, \chi, \kappa_h)^T. \tag{1}$$

More complex wind distributions over the rotor disk might be modelled using higher order terms. However, such local fluctuations would manifest themselves in higher Rev harmonics, complicating the estimation process.

The wind states are defined with respect to a nacelle-attached frame of reference with origin at the hub, made of three mutually orthogonal unit vectors $\boldsymbol{x}$, $\boldsymbol{y}$ and $\boldsymbol{z}$. The $\boldsymbol{x}$ vector is parallel to the rotor axis and pointing downwind, $\boldsymbol{z}$ points upward in the vertical plane, while $\boldsymbol{y}$ is defined according to the right-hand rule. The wind vector $\boldsymbol{V}$ is expressed in terms of its components in the nacelle frame as $\boldsymbol{V} = (u, v, z)^T$. The wind speed at the rotor disk $W(y,z) = |\boldsymbol{V}|$ is readily computed as

$$W(y,z) = V\left(\left(\frac{H+z}{H}\right)^{\kappa_v} + \frac{y}{R}\kappa_h\right), \tag{2}$$

where $V$ is the wind speed at hub height $H$, while $R$ is the rotor radius. The three wind velocity vector components are then expressed as

$$u(y,z) = W(y,z)\cos(\phi)\cos(\chi), \tag{3a}$$
$$v(y,z) = W(y,z)\sin(\phi)\cos(\chi), \tag{3b}$$
$$w(y,z) = W(y,z)\sin(\chi). \tag{3c}$$

Notice that, because of the definition of the nacelle-attached reference frame $(\boldsymbol{x}, \boldsymbol{y}, \boldsymbol{z})$, a horizontal wind results in an upflow equal to the negative of the nacelle uptilt angle. This is useful for separating the effects of gravitational loads from aerodynamic ones, as shown later on. To ease the interpretation of the results, all computed wind states reported in the numerical examples of the rest of this paper were mapped to a frame of reference similar to the nacelle-attached one, but whose $\boldsymbol{x}$ unit vector is horizontal with respect to the ground instead of being aligned with the rotor axis. Figure 1 illustrates the meaning of the four

wind states.

Two different wind fields are considered in the following. In the fully-parameterized case, the wind field is completely defined at each instant of time by $V$ and $\boldsymbol{\theta}$. On the other hand, a more realistic wind field is generated using the Kaimal




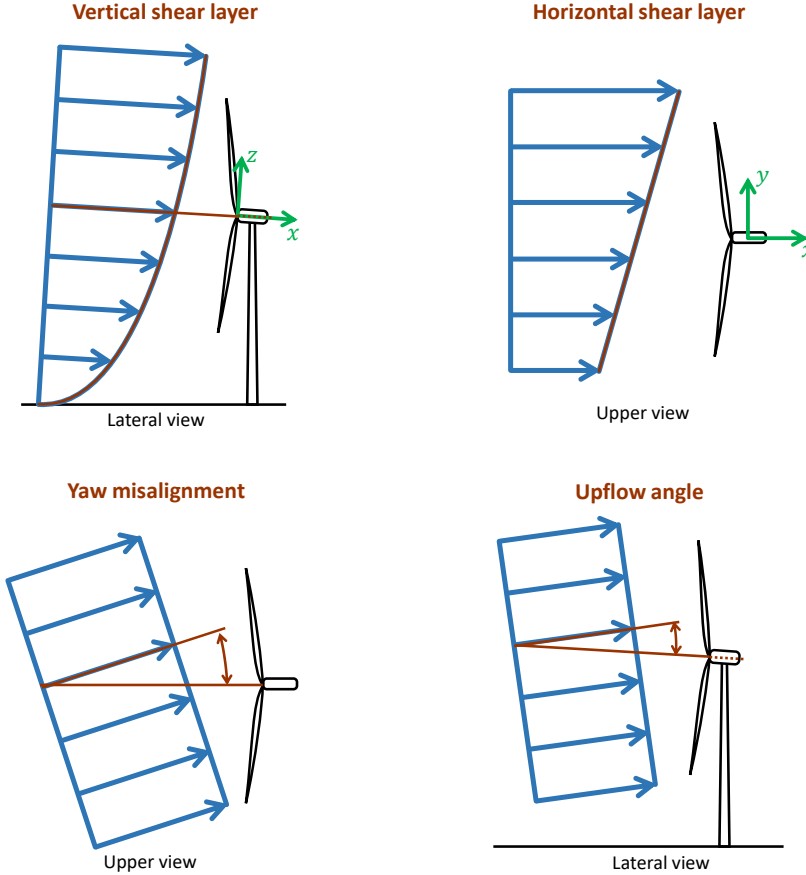

**Figure 1.** Definition of the four wind states used for parameterizing the wind field over the rotor disk.

turbulent wind model implemented in the open-source code `TurbSim` (Jonkman and Kilcher, 2012). In the latter case, the wind field can be considered as the superposition of a fully-parameterized wind with turbulent fluctuations possessing specific space-time characteristics. Given a wind turbine operating in a turbulent wind field, goal of the proposed observer is then to estimate online a wind state $\boldsymbol{\theta}$ that approximates the turbulent wind at each instant of time.

5 **2.2 Blade load harmonics**

Under the effects of a steady anisotropic wind, the response of a stable wind turbine converges to a periodic motion. In such a regime, a generic blade load $m$ can be expanded in Fourier series as

$$m(\psi) = m_0 + \sum_{n=1}^{\infty} \left( m_{nc} \cos(n\psi) + m_{ns} \sin(n\psi) \right), \tag{4}$$





where $\psi$ is the azimuth angle, subscripts $(\cdot)_{nc}$ and $(\cdot)_{ns}$ refer to the $n\times$Rev cosine and sine components, respectively, whereas $m_0$ is the 0th harmonic constant amplitude. For convenience, signal harmonics are collected in a vector

$$\boldsymbol{h} = (m_0, m_{1c}, m_{1s}, m_{2c}, m_{2s}, \ldots)^T, \tag{5}$$

which can be computed by demodulating the blade load signal $m(\psi)$ or, for rotors with at least three blades, by using the Coleman Feingold (or multi-blade coordinate) transformation (Coleman and Feingold, 1958; Bottasso and Riboldi, 2014). By using the latter method, harmonics at the $n\times$Rev frequency can be computed as

$$\left\{ \begin{array}{c} m_{nc} \\ m_{ns} \end{array} \right\} = \frac{2}{3} \left[ \begin{array}{ccc} \cos(n\psi_{(1)}) & \cos(n\psi_{(2)}) & \cos(n\psi_{(3)}) \\ \sin(n\psi_{(1)}) & \sin(n\psi_{(2)}) & \sin(n\psi_{(3)}) \end{array} \right] \left\{ \begin{array}{c} m_{(1)} \\ m_{(2)} \\ m_{(3)} \end{array} \right\}, \tag{6}$$

where $m_{(i)}$ and $\psi_{(i)}$ are the $i$th blade moment and azimuth angle, respectively. Similar relationships exist also for a higher number of blades, but not for a smaller one. It can be shown that this way harmonics at the $i\times$Rev are transformed into $0\times$Rev components, whereas the other harmonics are either canceled out or transformed into multiples of the number $B$ of blades. This implies that it is always necessary to filter around and above the $B\times$Rev frequency after having applied the Coleman transformation. Adaptive filtering can be used to follow the rotor speed changes caused by variations in the wind speed.

Both out-of-plane (superscript $(\cdot)^{\mathrm{OP}}$) and in-plane (superscript $(\cdot)^{\mathrm{IP}}$) blade bending harmonic components up to a desired Rev frequency are considered and collected in a vector $\boldsymbol{m}$, defined as

$$\boldsymbol{m} = \left( m_{1c}^{\mathrm{OP}}, m_{1s}^{\mathrm{OP}}, m_{1c}^{\mathrm{IP}}, m_{1s}^{\mathrm{IP}}, m_{2c}^{\mathrm{OP}}, m_{2s}^{\mathrm{OP}}, m_{2c}^{\mathrm{IP}}, m_{2s}^{\mathrm{IP}}, \ldots \right)^T. \tag{7}$$

### 2.3 Wind state observer

#### 2.3.1 Modeling of the load-wind relationship

The formulation of a wind state observer necessitates of a model expressing the dependency of the loads on the wind conditions, and in particular of the load harmonics $\boldsymbol{m}$ on the wind state vector $\boldsymbol{\theta}$. To this end, consider first a wind turbine model expressed by a set on nonlinear differential equations together with their output relations:

$$\boldsymbol{f}\big(\boldsymbol{x}, \dot{\boldsymbol{x}}, \boldsymbol{u}(\boldsymbol{\theta}, V, \varrho)\big) = \boldsymbol{0}, \tag{8a}$$

$$\boldsymbol{y} = \boldsymbol{g}\big(\boldsymbol{x}, \dot{\boldsymbol{x}}, \boldsymbol{u}(\boldsymbol{\theta}, V, \varrho)\big), \tag{8b}$$

where $\boldsymbol{x}$ is the state vector, $\boldsymbol{u}$ the input vector, whereas $\boldsymbol{y}$ indicates the output vector (containing, in this case, the blade bending moments). The input vector only includes the exogenous disturbance represented by the wind parameters $\boldsymbol{\theta}$, by the wind speed $V$ and the air density $\varrho$, because the presence of a feedback controller (usually in the form of a pitch-torque controller) can be considered to be included in the definition of the system model $\boldsymbol{f}(\cdot)$.



Under a steady input $\boldsymbol{u}$, the response of system (8a) in terms of its states converges to a periodic solution, which can be described through a truncated Fourier expansion as

$$\boldsymbol{x} = \boldsymbol{x}_0 + \sum_{n=1}^{N} \left( \boldsymbol{x}_{nc} \cos(n\psi) + \boldsymbol{x}_{ns} \sin(n\psi) \right). \tag{9}$$

Inserting (9) into (8a) and collecting all terms at the same frequency (a procedure termed *harmonic balance*), one can compute

$\boldsymbol{x}_{nc}$ and $\boldsymbol{x}_{ns}$, which clearly will depend on $\boldsymbol{\theta}$, $V$ and $\varrho$. Finally, the harmonics $\boldsymbol{x}_{nc}$ and $\boldsymbol{x}_{ns}$ can be inserted into the output Eq. (8b), yielding the desired relationship between load harmonics and wind parameters:

$$\boldsymbol{m} = \boldsymbol{\mathcal{M}}(\boldsymbol{\theta}, V, \varrho). \tag{10}$$

An example of this derivation for a simplified flapping blade model can be found in Bottasso and Riboldi (2014). In principle, the resulting input-output relationship should also include the dependency on other parameters, such as blade pitch and rotor

speed, as shown for example in Simley and Pao (2014). However, all these quantities depend in turn on the environmental and operating conditions according to the particular regulation strategy adopted by the on-board controller. Therefore, in this work the model is assumed to depend only on $\boldsymbol{\theta}$, $V$ and $\varrho$. Vector $\boldsymbol{\theta}$ is to be estimated with the proposed observer, while $V$, which is a scheduling parameter for the model, can be either measured or observed using a rotor-equivalent wind speed estimator (Soltani et al., 2013; Simley and Pao, 2014; Bottasso et al., 2015, 2016).

It is important to emphasize that this approach, which leads to a *white box model*, may suffer from inaccuracies. In fact any mismatch between model (8) and the reality will inevitably pollute the input-output relationship (10). To address this problem, one may calibrate some of the parameters of model (8) based on available measurements. This procedure, carried out using parameter identification techniques (Jategaonkar, 2006), leads to a *gray box model*.

In this paper, a third approach is used, which is entirely based on system identification. In this case, the desired input-output

relationship between loads and wind states is considered as a *black box*, which is identified directly from measurements of $\boldsymbol{m}$ and $\boldsymbol{\theta}$. This way, the need for an analytical model is bypassed completely. Clearly, the model structure has to be simple enough to be easily identified, but at the same time it should be able to describe the input-output relationship with the necessary level of precision. The advantage of avoiding the use of a white or gray model is paid in terms of the need for a set of measurements that is rich and complete enough to enable the identification of the relationship of interest.

The data set for the identification of the black box model can be obtained either by simulation or by measurements performed in the field. The former approach, which is also the one that was used for the present work, is relatively simple, because in fact in a simulation environment one can readily measure all necessary quantities (loads and wind states). In contrast to this simplicity, it is clear that here again any mismatch of the simulation model with respect to reality will affect the quality of the identified input-output model. While this is in principle a possible drawback, one should not forget that the present approach

only uses the very lowest harmonics (typically only the $1\times$Rev) of the response. State-of-the-art aeroservoelastic codes used for the design and certification of wind turbines are typically quite accurate in this frequency range, and many of these codes have been successfully validated in their ability to provide good quality estimates of the loads when compared with experimental data. An additional remark on this modeling approach is in order: it is clear that identifying a black box model based on the





outputs of a simulation is in a sense akin to the extraction of a white box model from the simulation model itself. However, given the level of complexity of modern comprehensive aeroservoelastic codes, the direct extraction of the necessary input-output relationship by manipulation of the underlying equations is hardly doable in practice, especially when working with legacy codes.

A second possible approach is to use field measurements. In this case the machine should be equipped with load sensors, as well as a met-mast, a LiDAR or other flow sensors to measure wind states. Each of these techniques implies its own hypothesis (e.g., frozen turbulence in the case of flow measurements performed away from the rotor disk), each is limited by its own specific inherent accuracy, and each is affected by errors and disturbances. While this approach is certainly possible and it was in fact successfully demonstrated in Bottasso and Riboldi (2014), it was not pursued further in the present work.

**2.3.2    Linear model**

Inspired by Eq. (10), a linear input-output model can be expressed as

$$\boldsymbol{m} = \boldsymbol{F}(V, \varrho)\boldsymbol{\theta} + \boldsymbol{m}_0(V, \varrho), \tag{11a}$$

$$= \boldsymbol{T}\overline{\boldsymbol{\theta}}, \tag{11b}$$

where $\boldsymbol{F}$ and $\boldsymbol{m}_0$ are the model coefficients, while $\boldsymbol{T}(V, \varrho) = [\boldsymbol{F}(V, \varrho), \, \boldsymbol{m}_0(V, \varrho)]$ and $\overline{\boldsymbol{\theta}} = (\boldsymbol{\theta}^T, \, 1)^T$.

Matrix $\boldsymbol{F}$ is the sensitivity of the harmonics with respect to the wind states and depends on the operating condition of the machine through the wind speed $V$ and the air density $\varrho$. Vector $\boldsymbol{m}_0$ is a term accounting for gravity-induced loads. In fact, when $\boldsymbol{\theta} = 0$, the wind field is a constant-over-the-rotor-disk flow parallel to the rotor axis, which only causes a 0×Rev load response and therefore it does not contribute to $\boldsymbol{m}$. Similarly, inertial effects due to the rotor spinning with an angular velocity $\Omega$ also generate only 0×Rev loads, and hence do not contribute to Eq. (11). Vector $\boldsymbol{m}_0$ can be expressed as

$$\boldsymbol{m}_0 = \boldsymbol{g} + qA\boldsymbol{c}(V, \varrho). \tag{12}$$

The first term, $\boldsymbol{g}$, accounts for in-plane and out-of-plane gravity-induced loads, the latter being caused by blade precone, prebend and rotor up-tilt. The second term, $qA\boldsymbol{c}$, is a gravity-induced load due to the rotor deformation caused by aerodynamic loads, which therefore can be nondimensionalized accordingly. For the same reasons noted above, also this term in general depends on $V$ and $\varrho$.

Separating the effects of gravity from aerodynamic-induced loads allows for the correction of air density changes. This is important in practise because density, being dependent on temperature, undergoes significant fluctuations in the field, thereby affecting load measurements. The split of gravity-induced terms in constant and aerodynamically-caused ones is also important, as it highlights the variability of the latter term with the operating condition.





The unknown matrix of coefficients $\boldsymbol{T}$ can be computed collecting multiple observations for the moments $\boldsymbol{m}^{(i)}$ and inputs $\overline{\boldsymbol{\theta}}^{(i)}$, where $(\cdot)^{(i)}$ indicates the $i$th of $N_{\text{exp}}$ available observations. Grouping the measurements in matrices

$$\boldsymbol{M} = \Big[\, \boldsymbol{m}^{(1)},\ \boldsymbol{m}^{(2)},\ \ldots,\ \boldsymbol{m}^{(N_{\text{exp}})}\,\Big], \tag{13a}$$

$$\boldsymbol{\Theta} = \Big[\, \overline{\boldsymbol{\theta}}^{(1)},\ \overline{\boldsymbol{\theta}}^{(2)},\ \ldots,\ \overline{\boldsymbol{\theta}}^{(N_{\text{exp}})}\,\Big], \tag{13b}$$

the input-output relationship (11) can be written collectively for all observations as

$$\boldsymbol{M} = \boldsymbol{T}\boldsymbol{\Theta}. \tag{14}$$

Finally, matrix $\boldsymbol{T}$ is readily estimated in a least-squares sense as

$$\boldsymbol{T} = \boldsymbol{M}\boldsymbol{\Theta}^T \left(\boldsymbol{\Theta}\boldsymbol{\Theta}^T\right)^{-1}. \tag{15}$$

The problem is solvable if and only if matrix $\boldsymbol{\Theta}$ has a full rank. In this sense, the condition number of matrix $\boldsymbol{\Theta}\boldsymbol{\Theta}^T$ gives an

indication of the identifiability of a model given a set of measurements. If the condition number is excessively high, then the problem is ill posed and the data set has to be enriched/modified.

As previously noted, the input-output model should be scheduled in terms of the wind speed $V$ and air density $\varrho$, as the model coefficients depend on the operating condition of the machine. To this end, a piece-wise linear scheduled model can be expressed as

$$\boldsymbol{m} = \sum_{k=1}^{N_{\text{node}_V}} \sum_{w=1}^{N_{\text{node}_\varrho}} \boldsymbol{F}_{k,w} n_{k,w}(V,\varrho)\boldsymbol{\theta} + \boldsymbol{m}_{0k,w} n_{k,w}(V,\varrho) = \sum_{k=1}^{N_{\text{node}_V}} \sum_{w=1}^{N_{\text{node}_\varrho}} \boldsymbol{T}_{k,w} n_{k,w}(V,\varrho)\overline{\boldsymbol{\theta}}, \tag{16}$$

where the wind speed and air density ranges have been discretized by introducing $N_{\text{node}_V}$ wind speed nodes and $N_{\text{node}_\varrho}$ density nodes, while $\boldsymbol{F}_{k,w}$ and $\boldsymbol{m}_{0k,w}$ are the model coefficient nodal matrices, grouped together as $\boldsymbol{T}_{k,w} = [\boldsymbol{F}_{k,w}\,\boldsymbol{m}_{0k,w}]$. Finally, two-dimensional shape functions are noted $n_{k,w}(V,\varrho)$. The scheduled model (16) can be written in a more compact form as

$$\boldsymbol{m} = \widehat{\boldsymbol{T}}\widehat{\boldsymbol{\theta}}(V,\varrho) \tag{17}$$

where $\widehat{\boldsymbol{\theta}}(V,\varrho) = \widehat{\boldsymbol{N}}(V,\varrho)\overline{\boldsymbol{\theta}}$ and

$$\widehat{\boldsymbol{T}} = \Big[\boldsymbol{T}_{1,1},\ \boldsymbol{T}_{1,2},\ \ldots,\ \boldsymbol{T}_{k,w},\ \ldots,\ \boldsymbol{T}_{N_{\text{node}_V},N_{\text{node}_\varrho}}\Big], \tag{18a}$$

$$\widehat{\boldsymbol{N}} = \Big[n_{1,1}(V,\varrho)\boldsymbol{I},\ n_{1,2}(V,\varrho)\boldsymbol{I},\ \ldots,\ n_{k,w}(V,\varrho)\boldsymbol{I},\ \ldots,\ n_{N_{\text{node}_V},N_{\text{node}_\varrho}}(V,\varrho)\boldsymbol{I}\Big]^T, \tag{18b}$$

being $\boldsymbol{I}$ an identity matrix of suitable dimensions.

Samples of the wind states and associated loads are now collected at $N_{\text{exp}}$ different operating conditions, each corresponding to its own wind speed $V^{(i)}$ and air density $\varrho^{(i)}$. The $i$th load vector and wind state vector are noted $\boldsymbol{m}^{(i)}$ and $\widehat{\boldsymbol{\theta}}(V^{(i)},\varrho^{(i)})$,




respectively. Both loads and wind states are collected into matrices as

$$\widehat{M} = \left[ \, \boldsymbol{m}^{(1)}, \; \boldsymbol{m}^{(2)}, \; \ldots, \; \boldsymbol{m}^{(N_{\exp})} \, \right], \tag{19a}$$

$$\widehat{\Theta} = \left[ \, \widehat{\boldsymbol{\theta}}(V^{(1)}, \varrho^{(1)}), \; \widehat{\boldsymbol{\theta}}(V^{(2)}, \varrho^{(2)}), \; \ldots, \; \widehat{\boldsymbol{\theta}}(V^{(N_{\exp})}, \varrho^{(N_{\exp})}) \, \right], \tag{19b}$$

leading to the overall system

$$\widehat{M} = \widehat{T} \widehat{\Theta}. \tag{20}$$

Finally, the matrix of unknown coefficients $\widehat{T}$ is computed in a least-squares sense as

$$\widehat{T} = \widehat{M} \widehat{\Theta}^T \left( \widehat{\Theta} \widehat{\Theta}^T \right)^{-1}. \tag{21}$$

The problem is well posed if the solving least-squares matrix, $\widehat{\Theta} \widehat{\Theta}^T$, is non-singular. One must clearly ensure that samples adequately cover all wind speed intervals, in order to ensure the identifiability of all nodal matrices $\boldsymbol{F}_{k,w}$ and $\boldsymbol{m}_{0k,w}$.

An example of the typical behavior of the model coefficients is given in Fig. 2, for the wind turbine described later on in this paper. The figure reports $\partial m_{1c}^{OP}/\partial \kappa_v$ (left) and $\partial m_{1s}^{IP}/\partial \chi$ (right) as functions of $V$ and $\varrho$. There is a distinctly different behavior with respect to wind speed of the load sensitivities in regions II (partial load) and III (full load). The rapid changes in the transition region $\text{II}\frac{1}{2}$ call for a suitable refinement of the node spacing in this regime. In general, the situation with respect to density is simpler, with small departures from a linear behavior only in the transition region.

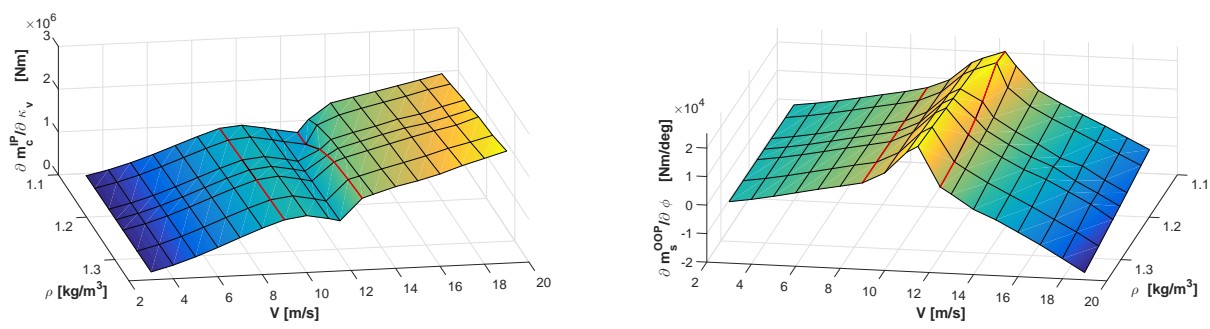

**Figure 2.** Behavior of two load sensitivities as functions of wind speed $V$ and air density $\varrho$: sensitivity of out-of-plane $1 \times \text{Rev}$ cosine moment with respect to vertical shear $\kappa_v$ (left) and in-plane $1 \times \text{Rev}$ sine moment with respect to upflow $\chi$ (right).

### 2.3.3 Nonlinear model

The assumption of linearity in the input-output relationship (10) might lead to inaccuracies. To correct for these potential effects while limiting model complexity, a model with an assumed degree of nonlinearity is formulated as

$$\boldsymbol{m} = \boldsymbol{F}_{NL} \boldsymbol{\theta}_{NL} + \boldsymbol{m}_{NL_0}. \tag{22}$$




The nonlinear wind state vector $\boldsymbol{\theta}_{\mathrm{NL}}$ contains, in addition to the elements of $\boldsymbol{\theta}$, also their nonlinear combinations $\theta_{\mathrm{NL}j}$ up to a given order $p$, where

$$\theta_{\mathrm{NL}j} = \prod_i \theta_i{}^{\alpha_i} \quad \text{s.t.} \quad \sum_i \alpha_i \leq p, \tag{23}$$

$\theta_i$ being the $i$th element of the linear wind state vector $\boldsymbol{\theta}$. For $p = 2$, which is the case considered here, the nonlinear wind state vector contains 14 terms:

$$\boldsymbol{\theta}_{NL} = (\phi, \kappa_v, \chi, \kappa_h, \phi\kappa_v, \phi\chi, \phi\kappa_h, \kappa_v\chi, \kappa_v\kappa_h, \chi\kappa_h, \phi^2, \kappa_v^2, \chi^2, \kappa_h^2)^T. \tag{24}$$

As the nonlinear model (22) is linear in the unknown coefficients $\boldsymbol{F}_{\mathrm{NL}}$ and $\boldsymbol{m}_{\mathrm{NL}_0}$, its identification is formally identical to the one of the linear model, both for the unscheduled and the scheduled cases. However, as more coefficients are present, one has to check here again that the data set is complete enough to guarantee the well posedness of the problem.

### 2.3.4 Wind turbine simulation model

In this work, an aeroservoelastic simulation model is used for representing the dynamic behavior of a wind turbine in all different scenarios of interest. The model represents a horizontal axis wind turbine with a rotor of 93 m of diameter with an uptilt of 4.5 deg, a hub height of 80 m and a rated power of 3 MW. The wind speeds at cut-in ($V_{\mathrm{CI}}$), rated power ($V_{\mathrm{RP}}$) and cut-out ($V_{\mathrm{CO}}$) are respectively equal to 3, 12.5 and 25 m/s. A rather wide transition region II$\frac{1}{2}$ extends from 9 to 12.5 m/s. The cut-in rotor speed is equal to 5.2 RPM, whereas the rated one is equal to 15 RPM. Both side-side and fore-aft tower frequencies $f_{\mathrm{tower}}$ are equal to 0.3 Hz. The first blade flap-wise frequency $f_{\mathrm{flap}}$ varies between 0.9 Hz at cut-in and 1 Hz at rated rotor speed. Finally, the first blade edge-wise mode $f_{\mathrm{edge}}$ is at around 1.5 Hz.

The aeroservoelastic model of the machine is developed using the finite element multibody code `Cp-Lambda` (Bauchau et al., 2003; Bottasso and Croce, 2006). The model includes flexible blades, tower and drive-train, implemented with geometrically exact nonlinear beam models (Bauchau, 2011). Rotor speed-dependent mechanical losses are considered within the drive-train-generator model, and compliant foundations are used to connect the tower base to the ground. The aerodynamics is rendered through the classical blade element momentum theory (BEM) and considers hub- and tip-losses, dynamic stall and unsteady corrections. The model is completed by an active pitch/torque controller, implemented as a speed-scheduled linear quadratic regulator (LQR) (Bottasso et al., 2012; Riboldi, 2012). Additionally, the pitch and torque actuators are modeled as second and first order systems, respectively. The total number of degrees of freedom included in the model is about 2500. Finally, the model is subjected to wind time histories generated by the code `TurbSim` (Jonkman and Kilcher, 2012).

### 2.3.5 Load-wind relationship in steady conditions

To test the performance of the linear and nonlinear models, the wind turbine was simulated in a variety of different operating conditions. Fully parameterized steady winds were generated at speeds $V = \{3, 4, 5, 6, 7, 8, 9, 11, 15, 19\}$ m/s, where for each




different wind speed all possible combinations of the following wind parameters were considered:

$$\phi = \{-16, -12, -8, -4, 0, 4, 8, 12, 16\} \text{ deg}, \tag{25a}$$

$$\kappa_v = \{0.0, 0.1, 0.2, 0.3, 0.4\}, \tag{25b}$$

$$\chi = \{0, 4, 8, 12\} \text{ deg}, \tag{25c}$$

$$\kappa_h = \{-0.1, -0.05, 0.0, 0.05, 0.1\}. \tag{25d}$$

Loads measured on the aeroelastic simulation model were decomposed in their harmonics at the 1×Rev and 2×Rev using the Coleman transformation and used, together with the corresponding wind states, for identifying linear and nonlinear models.

From the full range of tests performed, Fig. 3 shows two representative examples at a wind speed of 7 m/s, illustrating the match between the measurements obtained on the wind turbine simulation model (taken as ground truth) and the outputs of

10 the identified models. The ground truth is reported with markers, the linear model with solid lines and the nonlinear one with dashed lines. Figure 3 shows on the left $m_{1c}^{\mathrm{OP}}$ as a function of $\phi$, for different values of $\kappa_v$ and for $\kappa_h = 0.0$ and $\chi = 4$ deg. On the right, Fig. 3 shows $m_{1s}^{\mathrm{OP}}$ as a function of $\chi$, for different values of $\kappa_h$ and for $\kappa_v = 0.0$ and $\phi = 0$ deg. Both moments are nondimensionalized with respect to their own maximum absolute values.

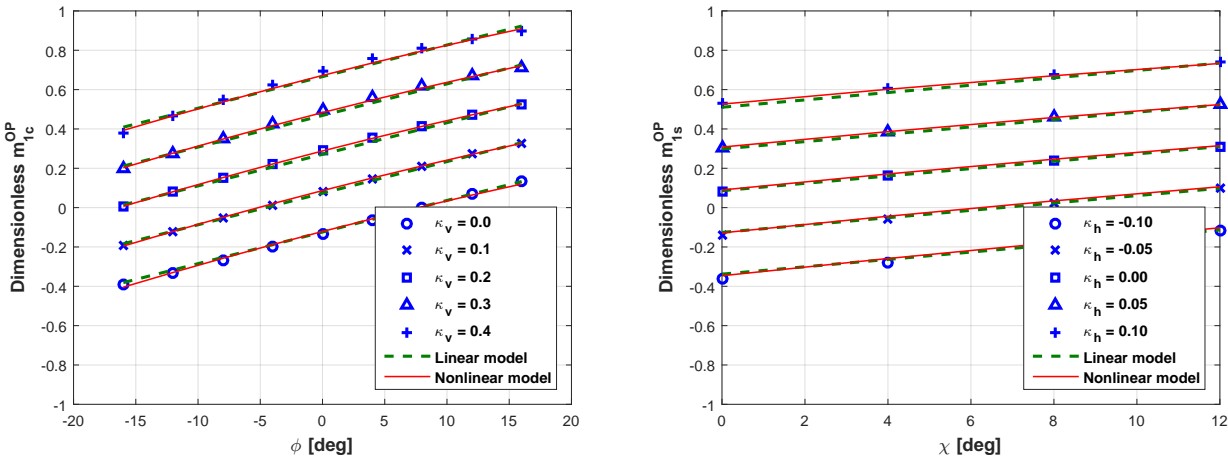

**Figure 3.** Comparison between measured and predicted harmonics, for the linear model (solid thick lines) and the nonlinear one of order 2 (solid thin lines). Normalized 1×Rev cosine (left) and sine (right) out-of-plane moment components are shown for different wind state variables.

The figures show that both models are capable of capturing the relevant behavior of the harmonics with respect to wind

states. The relationships appear to be linear, with only very minor nonlinearities. These analyses also graphically illustrate the sensitivity of the loads with respect to the wind parameters. As expected, even though all parameters have a certain effect on all loads, cosine harmonics are mainly influenced by the couple $\{\phi, \kappa_v\}$, whereas sine harmonics by $\{\chi, \kappa_h\}$. Similar considerations can be derived for the in-plane harmonics, not shown here for the sake of brevity.





On the other hand, the 2×Rev harmonics have a markedly different behavior, as shown in Fig. 4. The plots report the nondimensional out-of-plane 2×Rev cosine term on the left, and the in-plane 2×Rev sine term on the right, as functions of $\phi$ and for varying $\kappa_v$, with $\kappa_h = 0.0$ and $\chi = 4$ deg. Given the clear nonlinearity of the relationships, only the nonlinear model is able to capture the correct trends of these higher harmonics with respect to the wind states.

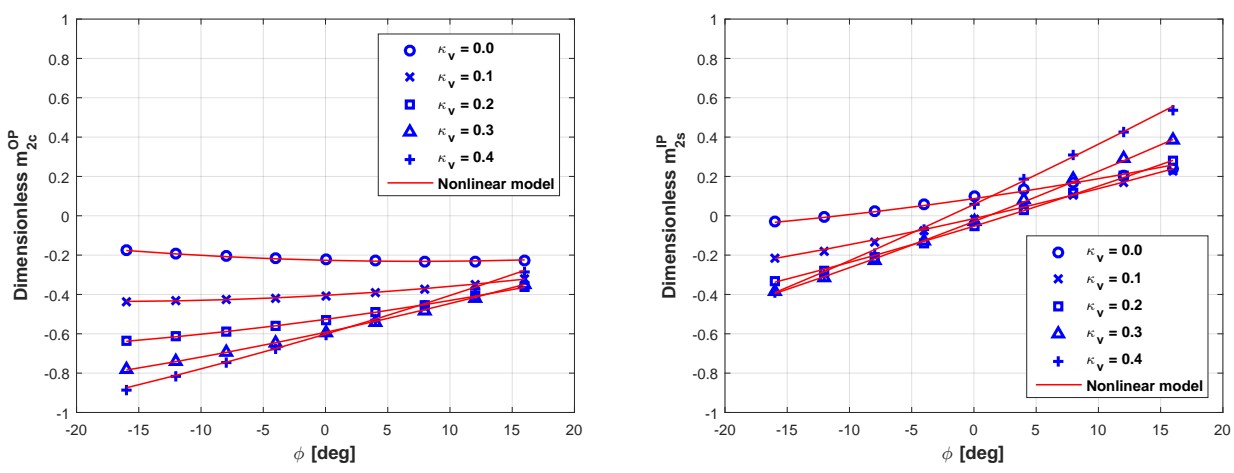

**Figure 4.** Comparison between measured and predicted harmonics for the nonlinear model of order 2 (solid thin lines). Normalized out-of-plane 2×Rev cosine (left) and in-plane 2×Rev sine (right) are shown for different wind state variables.

### 2.3.6 Choosing the number of harmonics

The previous analysis performed in steady wind conditions has shown that the 1×Rev harmonics exhibit a largely linear behavior with respect to the wind states, while the 2×Revs exhibit marked nonlinearities. In order to understand the behavior of the models in more realistic conditions, simulations were conducted in turbulent winds. In particular, it is necessary to establish whether the unsteadiness in the excitation provided by a turbulent wind is compatible with the steady-state harmonic models considered herein. In addition, as previously noted, a turbulent wind field cannot in general be exactly represented by the reduced set of wind states.

To investigate these effects, a 10 minute simulation was performed at 5 m/s mean wind speed with a TI equal to 20%, and null mean yaw misalignment, upflow, vertical and horizontal shears. At each instant of time, values of the wind parameters were computed from the wind grid generated with `TurbSim` (Jonkman and Kilcher, 2012) by fitting in a least-squares sense Eqs. (2) and (3). Blade load harmonics were extracted from the simulated outputs using the Coleman transformation and filtered with a low-pass 6$^\text{th}$-order Butterworth filter with a cut-out frequency of 0.14 Hz, in order to remove the remaining 3×Rev harmonic content in the Coleman-transformed moments. Figure 5 shows a comparison of the harmonics extracted from the simulation (shown in a thick blue solid line, and here again assumed to represent the ground truth) with those predicted by the second





order nonlinear model (shown using a thin red solid line), fed with the wind parameters computed from the wind grid. The plot on the left shows moment $m_{1c}^{IP}$, while the one on the right moment $m_{2c}^{OP}$.

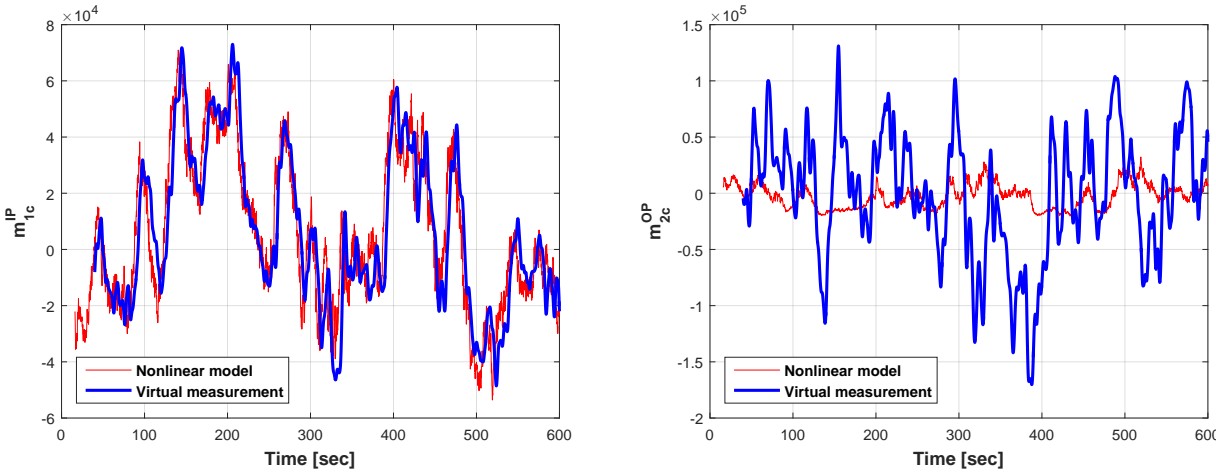

**Figure 5.** Comparison in turbulent wind conditions between measured harmonics (thick solid line) and harmonics predicted with a second order nonlinear model (thin solid line). Left: in-plane $1{\times}$Rev cosine component; right: out-of-plane $2{\times}$Rev cosine component.

By looking at the left plot of Fig. 5, it appears that there is an excellent match between predictions and measurements for the in-plane $1{\times}$Rev cosine harmonic. The small delay between the two signals is due to the filter used for removing higher frequencies. Both linear and nonlinear models yield similarly accurate results also for the sine and out-of-plane components, not reported here for brevity. These results show that, by and large, $1{\times}$Rev harmonics are primarily influenced by the wind states used here for parameterizing the wind field, with only small disturbances caused by turbulent fluctuations and blade dynamic effects. In this sense, $1{\times}$Rev harmonics are good candidates for feeding a wind state observer.

On the other hand, the right plot of the same figure shows a completely different behavior of measurements and predictions for the $2{\times}$Rev components in turbulent conditions. It should be remarked that, as previously illustrated in Fig. 4, the model is perfectly capable of capturing with good accuracy these higher harmonics in steady wind conditions. The reason for the very poor results of the turbulent case is due to the fact that small scale turbulent fluctuations in the wind field cause $2{\times}$Rev harmonics that are comparable to, if not larger than, the ones caused by the wind states used for the parameterization. Therefore, although $2{\times}$Rev harmonics carry information on the wind states, this information cannot be separated from the pollution brought by the smaller-scale wind field fluctuations. In this sense, $2{\times}$Rev harmonics are not good candidates for the observation of wind states. Based on these results, in the continuation of this work the vector of blade harmonics is limited to the $1{\times}$Revs and it is simply defined as

$$\boldsymbol{m} = \left( m_{1c}^{OP}, m_{1s}^{OP}, m_{1c}^{IP}, m_{1s}^{IP} \right)^T . \tag{26}$$





### 2.3.7 Wind state estimation

The problem of computing an estimate $\boldsymbol{\theta}_{\mathrm{E}}$ of the wind state vector given a load harmonic vector $\boldsymbol{m}_{\mathrm{M}}$ is considered next. Given the input-output model (10), a measured load $\boldsymbol{m}_{\mathrm{M}}$ can be expressed as

$$\boldsymbol{m}_{\mathrm{M}} = \mathcal{M}(\boldsymbol{\theta}, V, \varrho) + \boldsymbol{r}, \tag{27}$$

where $\boldsymbol{r}$ is the measurement error with covariance $\boldsymbol{R} = \boldsymbol{E}\left[\boldsymbol{r}\boldsymbol{r}^T\right]$. The residual is assumed to be zero-mean, white and Gaussian. The residual is due not only to measurement noise, but also to all effects not captured by the model, such as sampling and discretization errors, unmodeled nonlinearities and turbulence-induced loads. This implies that the assumption of a zero-mean, white and gaussian noise can be far from real.

The generalized least-squares estimate of $\boldsymbol{\theta}$ given $\boldsymbol{m}_{\mathrm{M}}$ is

$$\boldsymbol{\theta}_{\mathrm{E}} = \arg\min_{\boldsymbol{\theta}} \left( \left(\boldsymbol{m}_{\mathrm{M}} - \mathcal{M}(\boldsymbol{\theta}, V, \varrho)\right)^T \boldsymbol{R}^{-1} \left(\boldsymbol{m}_{\mathrm{M}} - \mathcal{M}(\boldsymbol{\theta}, V, \varrho)\right) \right). \tag{28}$$

Consider now linear model (11a) and assume $V$ to be known. The solution of problem (28) can be worked out analytically as

$$\boldsymbol{\theta}_{\mathrm{E}} = \left(\boldsymbol{F}(V)^T \boldsymbol{R}^{-1} \boldsymbol{F}(V)\right)^{-1} \boldsymbol{F}(V)^T \boldsymbol{R}^{-1} (\boldsymbol{m}_{\mathrm{M}} - \boldsymbol{m}_0). \tag{29}$$

Vector $\boldsymbol{\theta}_{\mathrm{E}}$ is *structurally identifiable* (or *observable*) if matrix $\boldsymbol{F}(V)^T \boldsymbol{R}^{-1} \boldsymbol{F}(V)$ is non singular. The structural identifiability analysis, which reveals when the estimation problem is well posed and with which accuracy it can be solved, will be analyzed 15   in Section 3.

For the nonlinear model (22), the solution of problem (28) involves a nonlinear unconstrained minimization, which was solved here starting from a suitable initial guess by the Levenberg-Marquardt method (More, 1977). As multiple local solutions may characterize the nonlinear problem, a global search algorithm or multiple starting points should be used for finding the optimum. Here again, one must verify observability, as discussed later in Section 3.

Estimator (28) was first characterized in steady wind conditions, and the results of this analysis are shown next. All plots are arranged in a similar way: any estimated wind state variable is plotted on the y-axis as a function of its corresponding ground truth quantity, reported on the x-axis. A black thin solid line indicates the bisector of the plot, representing a perfect match between the two quantities. Estimates are plotted using markers and thick solid lines for different wind conditions. Clearly, any deviation from the bisector directly indicates an estimation error.

Figure 6 shows an excerpt of the results obtained with the linear model for different wind conditions at 5 m/s. The estimates appear to be of good accuracy for all wind state variables, although some small errors affect the two angles. The reason for this behavior can be traced back to mild nonlinearities, clearly not captured by the linear model, that affect to a greater extent angles than shears. Among the wind parameters, the upflow seems to be the least accurate, while the horizontal shear appears as the most precise. Similar results, not shown here, were obtained for different wind speeds and flow conditions.

The matching improves with the use of the nonlinear model, as reported in Fig. 7. The plots show that all quantities appear to be well captured, with a clear improvement in the quality of the results.




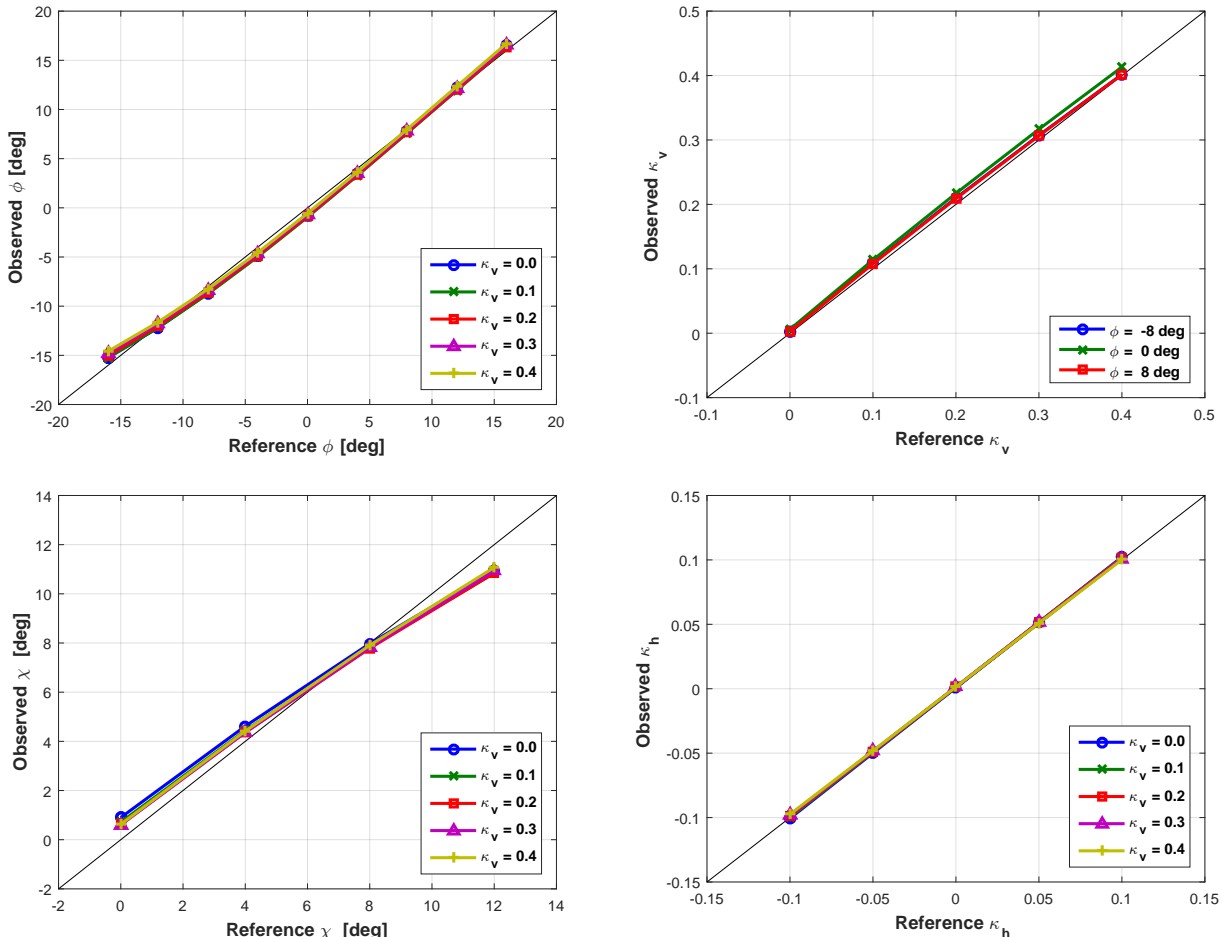

**Figure 6.** Wind states observed using the linear model for different steady inflow conditions at 5 m/s: yaw misalignment $\phi$ at $\chi = 8$ deg and $\kappa_h = -0.1$ (top left), vertical shear $\kappa_v$ at $\chi = 8$ deg and $\kappa_h = -0.1$ (top right), upflow angle $\chi$ at $\phi = -8$ deg and $\kappa_h = -0.1$ (bottom left), horizontal shear $\kappa_h$ at $\chi = 8$ deg and $\phi = -8$ deg (bottom right).




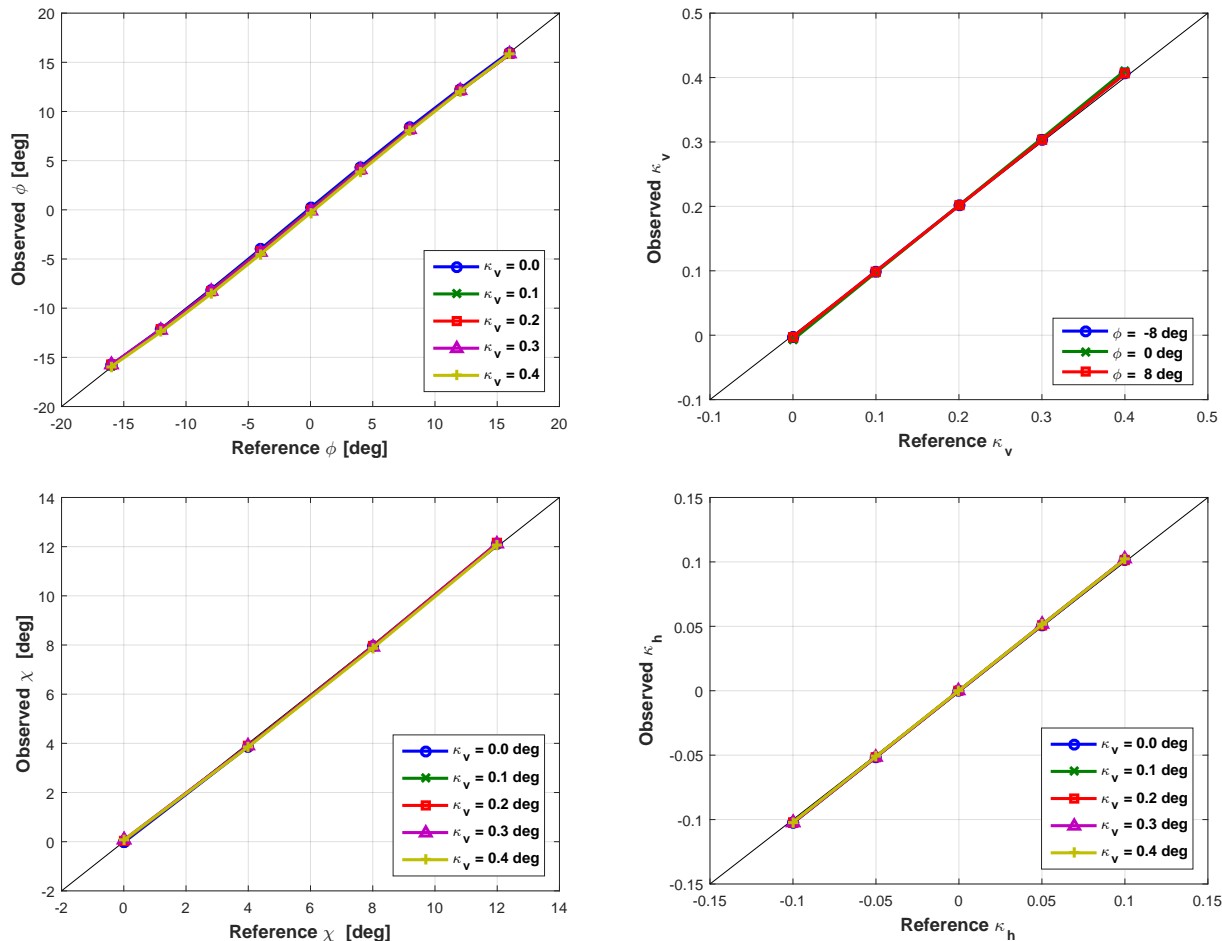

**Figure 7.** Wind state observed with the nonlinear model for different steady inflow conditions at 5 m/s: yaw misalignment $\phi$ at $\chi = 8$ deg and $\kappa_h = -0.1$ (top left), vertical shear $\kappa_v$ at $\chi = 8$ deg and $\kappa_h = -0.1$ (top right), upflow angle $\chi$ at $\phi = -8$ deg and $\kappa_h = -0.1$ (bottom left), horizontal shear $\kappa_h$ at $\chi = 8$ deg and $\phi = -8$ deg (bottom right).





## 3  A priori observability analysis

The observability of the wind parameters is analyzed next. As one can easily imagine, the level of accuracy of the estimates strongly depends on the sensitivity of the moments with respect to the to-be-estimated parameters and to the noise in the measurements.

Assuming a linear model, the real (unknown) wind state vector $\boldsymbol{\theta}_\mathrm{R}$ is related to the measured load vector $\boldsymbol{m}_\mathrm{M}$ as

$$\boldsymbol{m}_\mathrm{M} = \boldsymbol{F}\boldsymbol{\theta}_\mathrm{R} + \boldsymbol{m}_0 + \boldsymbol{r}. \tag{30}$$

Inserting (30) into (29), the estimation error $\boldsymbol{\epsilon_\theta}$ is readily derived as

$$\boldsymbol{\epsilon_\theta} = \boldsymbol{\theta}_\mathrm{E} - \boldsymbol{\theta}_\mathrm{R} = \left(\boldsymbol{F}^T \boldsymbol{R}^{-1} \boldsymbol{F}\right)^{-1} \boldsymbol{F}^T \boldsymbol{R}^{-1} \boldsymbol{r}. \tag{31}$$

The estimate is *unbiased*, as in fact the expected value of the error $\boldsymbol{E}\left[\boldsymbol{\epsilon_\theta}\right]$ is equal to zero when the residual is zero-mean.

Additionally, the covariance of the estimation error $\mathrm{Cov}\left[\boldsymbol{\epsilon_\theta}\right] = \mathcal{E}\left[\boldsymbol{\epsilon_\theta}\boldsymbol{\epsilon_\theta}^T\right]$ (Cramer, 1946) writes

$$\mathrm{Cov}\left[\boldsymbol{\epsilon_\theta}\right] = \left(\boldsymbol{F}^T \boldsymbol{R}^{-1} \boldsymbol{F}\right)^{-1}. \tag{32}$$

This expression shows the interplay between noise $\boldsymbol{r}$ and sensitivity $\boldsymbol{F}$, captured by the term $\boldsymbol{R}^{-\frac{1}{2}}\boldsymbol{F}$: the higher the variance and/or the lower the sensitivity of the measurements with respect to the wind states, the worst the accuracy of the estimates.

The covariance $\mathrm{Cov}\left[\boldsymbol{\epsilon_\theta}\right]$ expressed by Eq. (32) is typically fully populated, as the errors of the estimates are correlated. To

ease the understanding of the estimation problem, the SVD (Golub and van Loan, 1996) can be used to decouple the estimates. In fact, matrix $\boldsymbol{R}^{-\frac{1}{2}}\boldsymbol{F}$ can be factored as

$$\boldsymbol{R}^{-\frac{1}{2}}\boldsymbol{F} = \boldsymbol{U}\boldsymbol{\Sigma}\boldsymbol{V}^T, \tag{33}$$

where $\boldsymbol{U} \in \mathfrak{R}^{m \times m}$, $\boldsymbol{\Sigma} \in \mathfrak{R}^{m \times n}$ and $\boldsymbol{V} \in \mathfrak{R}^{n \times n}$, being $m$ the number of measurements and $n$ the number of wind state variables. Matrices $\boldsymbol{U}$ and $\boldsymbol{V}$ are orthonormal, i.e. $\boldsymbol{U}^T\boldsymbol{U} = \boldsymbol{U}\boldsymbol{U}^T = \boldsymbol{I}$ and $\boldsymbol{V}^T\boldsymbol{V} = \boldsymbol{V}\boldsymbol{V}^T = \boldsymbol{I}$, whereas $\boldsymbol{\Sigma} = \mathrm{diag}(\dots, 1/\sigma_i, \dots)$

is a diagonal matrix and $\sigma_i$ the standard deviation. Inserting Eq. (33) into Eq. (32), the covariance of the estimation error can be expressed as

$$\mathrm{Cov}\left[\boldsymbol{V}^T\boldsymbol{\epsilon_\theta}\right] = \boldsymbol{E}\left[\left(\boldsymbol{V}^T\left(\boldsymbol{\theta}_\mathrm{E} - \boldsymbol{\theta}_\mathrm{R}\right)\right)\left(\boldsymbol{V}^T\left(\boldsymbol{\theta}_\mathrm{E} - \boldsymbol{\theta}_\mathrm{R}\right)\right)^T\right] = \left(\boldsymbol{\Sigma}^T\boldsymbol{\Sigma}\right)^{-1} = \mathrm{diag}(\dots, \sigma_i^2, \dots). \tag{34}$$

This way, the problem is reformulated by the change of variables $\boldsymbol{\xi} = \boldsymbol{V}^T\boldsymbol{\theta}$, where $\boldsymbol{\xi}$ are statistically independent variables with diagonal covariance. This reformulation simplifies the interpretation of the structural observability of the problem. In fact,

the $i$th column of matrix $\boldsymbol{V}$ linearly combines the wind parameters, mapping them into a new parameter $\xi_i$ with variance $\sigma_i^2$. Clearly, a high variance indicates a low level of identifiability of the associated linear combination of wind parameters.

This analysis also provides information on the dependence of loads on wind states. In fact, one can easily show that

$$\frac{\partial \boldsymbol{m}}{\partial \boldsymbol{\xi}} = \boldsymbol{R}^{\frac{1}{2}}\boldsymbol{U}\boldsymbol{\Sigma}. \tag{35}$$





Therefore, the analysis of $U$ reveals on which linear combination of inflow parameters each load depends the most.

The same analysis can be applied to the nonlinear case, by linearizing Eq. (22) around a specific operating and wind condition and using $F = \partial(F_{NL}\theta_{NL})/\partial\theta = F_{NL}\partial\theta_{NL}/\partial\theta$.

### 3.1  Results of the a priori analysis

The a priori analysis was applied to the identified input-output models. Three different values of the noise covariance $R$ were considered. In the first two cases, all measures were supposed to be uncorrelated and affected by the same noise level, i.e. $R = \gamma^2 I$, where $\gamma$ is a positive real number. In the first case, $\gamma$ was set equal to $0.01 m_{\min}$, being $m_{\min}$ the minimum of the load amplitude maxima. In the second case, $\gamma$ was set to $0.01 m_{\max}$, being $m_{\max}$ the maximum of the load amplitude maxima. In the third case, the noise covariance was computed based on the differences between the loads $m_{obs}$ obtained by using the
observation model and the ones measured on the simulation model, $m_{sim}$, i.e.

$$R_\epsilon = \frac{1}{N_{exp}} \sum_{i=1}^{N_{exp}} (m_{obs\,i} - m_{sim\,i})(m_{obs\,i} - m_{sim\,i})^T. \tag{36}$$

For the first case, matrices $V$ and $U$ were computed at a wind speed of 7 m/s, obtaining

$$V = \begin{bmatrix} \sim 0 & \sim 0 & \mathbf{0.55} & \mathbf{0.83} \\ \sim 0 & \mathbf{\sim 1} & \sim 0 & \sim 0 \\ \sim 0 & \sim 0 & \mathbf{0.83} & \mathbf{0.55} \\ \mathbf{\sim 1} & \sim 0 & \sim 0 & \sim 0 \end{bmatrix}, \qquad U = \begin{bmatrix} 0.11 & 0.97 & 0.14 & 0.18 \\ -0.97 & 0.11 & 0.18 & -0.14 \\ 0.03 & 0.22 & -0.60 & -0.77 \\ -0.23 & 0.02 & -0.77 & 0.60 \end{bmatrix}. \tag{37}$$

To interpret these results, remember that the wind state vector is defined as $\theta = (\phi, \kappa_v, \chi, \kappa_h)^T$, whereas the load vector as
$m = (m_{1c}^{OP}, m_{1s}^{OP}, m_{1c}^{IP}, m_{1s}^{IP})^T$.

The first and second columns of $V$ are related to the horizontal and vertical shears, respectively. Since their maximum entries approach 1, both parameters can be independently identified. On the other hand, a coupling between the two angles can be noticed from the third and fourth column: an error in the estimation of one angle will propagate and affect the estimate of the other. Similar $V$ matrices, leading to the same conclusions, were computed at different wind speeds and different noise
levels $\gamma^2$.

To interpret matrix $U$, consider that rows are associated with entries of the load vector, whereas columns with entries of the wind state vector. The first column of $U$ shows that the horizontal shear mostly affects the sine components of both out- and in-plane moments. Similarly, the second column shows that the vertical shear mostly affects the cosine components of the loads. On the other hand, the third and fourth columns, associated with the angles, do not indicate a predominant effect
on some load components. In fact, all loads are affected by both upflow and yaw misalignment, with the in-plane harmonics exhibiting a bit higher sensitivity.

As a side observation, notice also the symmetry between the couples $\{\phi, \kappa_v\}$ and $\{\chi, \kappa_h\}$, an effect of the near 90 deg-symmetry in the definition of the wind parameters and in the response of the machine (see Fig. 1).



Table 1 reports the expected variances of the wind state estimation errors for the three considered noise variances. It appears that, as expected, higher noise levels are associated with higher variances of the estimates. In addition, the variance of the angles appears to be significantly higher than the one of the shears. In fact, angle variances approach and exceed tens of degrees for the higher noise levels, indicating that instantaneous estimates of these wind states are probably impractical. However, longer

5  term observation could be possible by time filtering, as discussed and shown later on.

**Table 1.** Expected variance of wind state estimates based on the a priori analysis.

| Standard deviations | $0.01\,m_{min}$ | $0.01\,m_{max}$ | $\boldsymbol{R}_\epsilon$ |
|---|---|---|---|
| $\sigma_\phi[\text{deg}]$ | 0.95 | 26.0 | 5.9 |
| $\sigma_{\kappa_v}$ | 1.1e-3 | 3.0e-2 | 3.3e-3 |
| $\sigma_\chi[\text{deg}]$ | 0.81 | 22.3 | 9.0 |
| $\sigma_{\kappa_h}$ | 1.1e-3 | 3.0e-2 | 3.3e-3 |

Finally, Fig. 8 shows the standard deviations (STD) of the wind parameter estimates with respect to the wind speed, computed asuming $\boldsymbol{R} = \gamma\boldsymbol{I}$, with $\gamma = 0.01(m_{min} + m_{max})/2$. The plot shows a marked improvement of the quality of the estimates with wind speed.

Similar results, not shown here for the sake of brevity, were obtained with the nonlinear model.

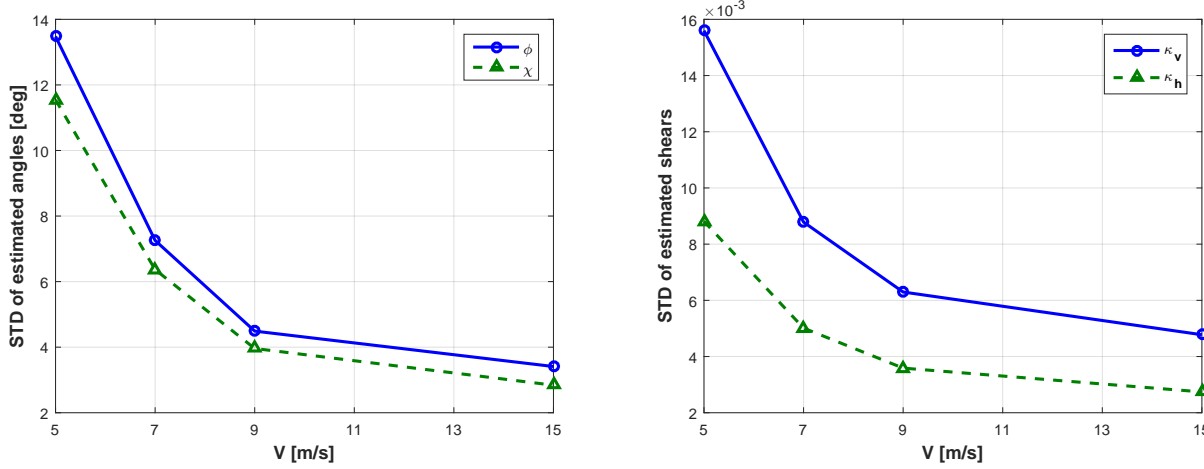

**Figure 8.** Expected standard deviation of the wind state estimates as function of wind speed. Left: standard deviations for angles $\phi$ and $\chi$; right: standard deviations for shears $\kappa_v$ and $\kappa_h$.





### 3.2 Expected observer behavior

Given the behavior of the linear and nonlinear observers and the results of the SVD-based a priori observability analysis, the following considerations can be made:

- In general it should be possible to estimate both shears with a satisfactory precision, as their errors are moderate even
for significant measurement noise levels.

- It is expected that the estimation of both yaw misalignment and upflow angle will be more significantly affected by measurement noise. Because of this, the estimation of these angles should be accompanied by a suitable filtering action in order to remove fast fluctuations. This also implies that these angles can only be estimated on longer time horizons than in the case of shears.

- The observation accuracy should increase with increasing wind speed.

- The nonlinear model appears to be more accurate than the linear one for the estimation of yaw misalignment and upflow angles. On the other hand, shears seem to be captured well also by the linear model.

The different expected accuracy in the estimation of shears and angles can be given an even more intuitive explanation. Consider in fact the blade section depicted in Fig. 9. The relative airflow velocity vector can be decomposed into the component
$V_\perp = (1-a)V$ perpendicular to the rotor disk plane, where $a$ is the local induction factor, and the one $V_{//} = \Omega r$ parallel to it, being $r$ the section radial position.

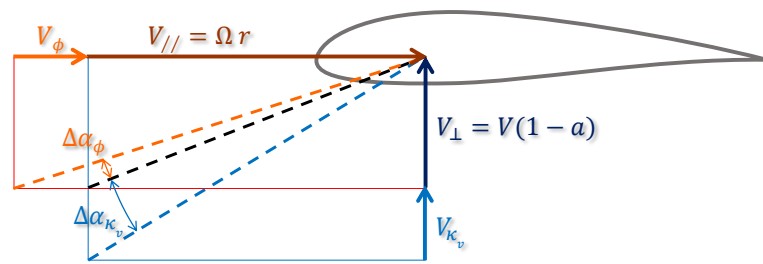

**Figure 9.** Effects of shear and misalignment changes on sectional angle of attack.

A change of shear will be seen by the blade section mainly as a change of $V_\perp$. On the other hand, a change in misalignment will induce a change mainly in $V_{//}$. The figure shows that two equal velocity perturbations $\Delta V = V_{\kappa_v} = V \sin(\phi)$, respectively perpendicular and parallel to the rotor plane, will induce different changes in the sectional angle of attack. In particular, the
change due to a perpendicular (shear-caused) variation is larger than the one due to a parallel (misalignment-caused) variation.



This is also easily shown by considering that the inflow angle is $\tan\zeta = V_\perp/V_{//}$. Hence, for a perturbation $\Delta V$ due to shear variation, the inflow changes as $\tan\zeta = (V_\perp + \Delta V)/V_{//}$. On the other hand, for a perturbation $\Delta V$ due to misalignment variation, the inflow changes as $\tan\zeta = V_\perp/(V_{//} + \Delta V)$. For typical values of $V_\perp$ and $V_{//}$, Fig. 10 shows the behavior of $\tan\zeta$ as a function of $\Delta V$. As clearly shown by the plot, for a same perturbation $\Delta V$ (say of 1 m/sec, as shown in the figure

by way of example), the ensuing change in inflow angle is larger when the perturbation is due to a change in shear than when it is due to a change in misalignment. This implies a similarly larger variation of the sectional angle of attack, and hence of the loads. In conclusions, one may expect that the rotor response will be more sensitive to variations in shear than in misalignment, when these two different phenomena produce velocity perturbations of the same magnitude. Due to the rotational symmetry of the problem, the same conclusions clearly hold true for a variation in horizontal shear, or for a variation of the vertical upflow

angle.

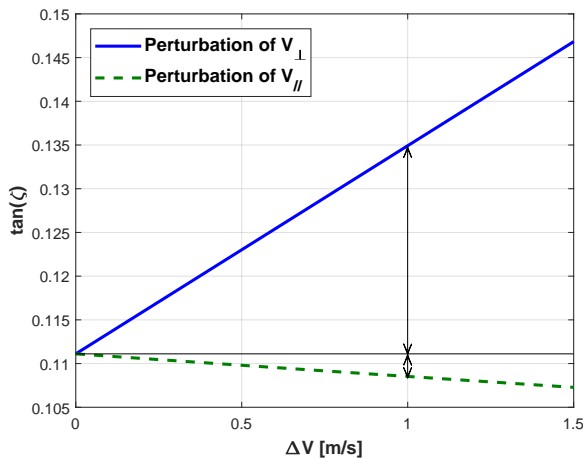

**Figure 10.** Variation of the inflow angle $\zeta$ at a blade section, as a function of a perturbation $\Delta V$ either in a direction perpendicular (solid line) or parallel (dashed line) to the rotor plane.

## 4   Results

After having verified in the previous sections that blade load harmonics carry enough information to infer wind states in steady conditions, attention is now turned to the dynamic problem. The non-turbulent case is considered first, using fully-parameterized wind fields with variable-in-time wind states. Next, the turbulent case is considered, using wind fields modeled

by the Kaimal method for different constant mean wind states. Finally, turbulent conditions with variable-in-time mean quantities are considered.




### 4.1 Non-turbulent case with fully-parameterized wind fields

Ideal non-turbulent and fully-parameterized wind fields with time-varying wind states were generated according to Eq. (3), by independently varying angles $\phi$ and $\chi$ as well as shears $\kappa_v$ and $\kappa_h$. Here and in the following examples, load harmonics were extracted from the simulated wind turbine response by using the Coleman transformation, followed by filtering with an 5 $8^{\text{th}}$-order Butterworth filter with cut-out frequency equal to $0.35 f_{\text{tower}} = 0.105$ Hz to remove load oscillations at the tower frequency. Finally, inflow conditions were estimated with the proposed observer and compared with the real ones.

Figure 11 shows the results obtained at 4 and 9 m/s, respectively in the left and right plots, using the linear and nonlinear models. The agreement is generally good as all parameters are well observed by both models. The observed states are affected by a delay of about 7 seconds, primarily due to the effects of the filter. There are minor differences between the linear and the 10 nonlinear models, which however are not large enough to allow drawing any conclusions.

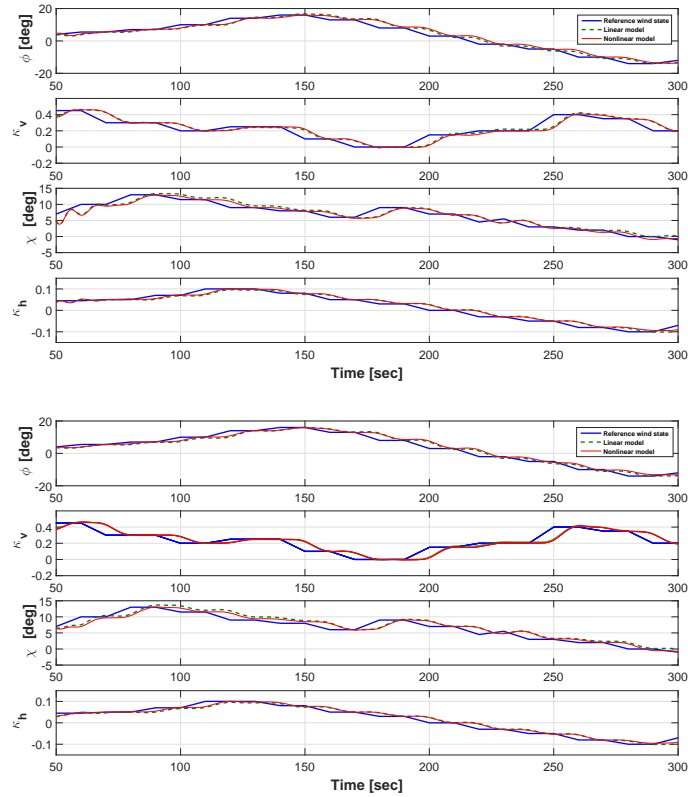

**Figure 11.** Wind state observations in non-turbulent wind conditions with variable wind parameters at 4 (left) and 9 m/s (right). Solid thick blue lines: real wind parameters; dashed thick green lines: observations by the linear model; solid thin red lines: observations by the nonlinear model.



## 4.2 Turbulent case

Different turbulent wind fields were generated using the `TurbSim` software according to the Kaimal model. The corresponding inflow conditions, in terms of hub-height wind speed $V$ and wind states $\boldsymbol{\theta}$, were then computed by fitting at each instant of time the wind state parametrization (1) to the turbulent wind grid over the complete rotor disk. The wind parameters obtained this way were then used as reference quantities to verify the accuracy of the estimated ones.

As wind states are inferred from blade loads, which in turn depend on the wind conditions at the location occupied by each single blade at each time instant, also an alternative way of computing the reference wind conditions was used. In this second implementation, wind parameters were computed by fitting the wind state parametrization expressed by Eq. (2) and Eq. (3) not over the complete rotor disk, but only to its portion occupied at that time instant by the three blades. Spanwise weighting was also used, on account of the non-uniform power extraction characteristics of rotors (Soltani et al., 2013). As the two methods do not yield significantly different reference wind states, only the results obtained with the first approach are reported in the following.

Figures 12 and 13 report the results obtained at 7 and 19 m/s, which belong respectively to regions II and III. Each figure shows on the left results for a TI equal to 2%, and on the right for the 12% TI case.

These results suggest several possible considerations.

First, the estimates of both shears $\kappa_v$ and $\kappa_h$ appear to have in general a high accuracy: their mean values as well as their rapid oscillations are well captured by both the linear and nonlinear models. Here again, results are affected by a 7 s delay induced by the filter. For the lower wind speeds and turbulence intensities, the linear and nonlinear observers exhibit a very similar behavior. However, differences appear at 19 m/s and 12% TI, as shown by Fig. 13 on the right. In fact, between second 250 and second 350 of the simulation, the estimation of the vertical shear provided by the linear model is affected by large errors, whereas the nonlinear observer results still remain acceptable.

The good behavior of the shear estimates suggests the possible use of a faster filter in order to reduce the estimation delay. For example, the delay can be reduced to only 4 s by using a filter cut-out frequency of 0.17 Hz, which corresponds to 1.2 times the rotor frequency at 5 m/s.

On the other hand, estimation of the angles $\phi$ and $\chi$ does not prove to be as accurate as the one of the shears, as fully expected based on the a priori observability study. Mean values are well captured, especially by the nonlinear model, but fluctuations are missed by both observers.

The general lower quality of the estimates for the angles was previously explained by the a priori analysis, and it is clearly illustrated a posteriori by the simulation results shown here. Various sources of error may ultimately be responsible for the oscillations in the estimates shown by the plots, including unmodeled dynamics, rapid pitch motions, or variable rotor speed. It is interesting to recall that the steady model (10) appeared well capable of capturing the behavior of the 1×Rev loads also in turbulent conditions, as clearly illustrated by the results shown in the left plot of Fig. 5. Notwithstanding this apparently more than satisfactory behavior when used to simulate loads given wind states, the inversion of the model to yield wind states given loads appears to be more problematic. In fact, because of the general lower level of observability of the angles with respect to




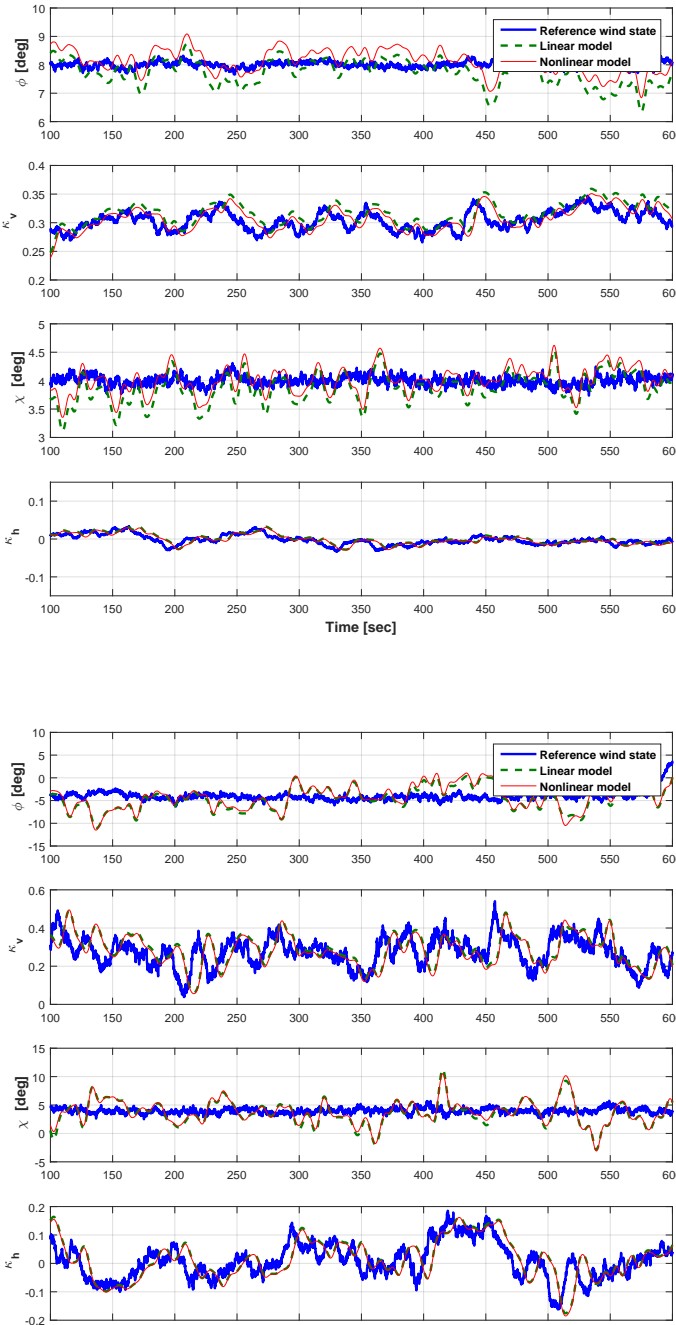

**Figure 12.** Wind state observations in turbulent wind conditions at 7 m/s for a TI equal to 2% (left) and 12% (right). Solid thick blue lines: reference wind parameters; dashed thick green lines: observations by the linear model; solid thin red lines: observations by the nonlinear model.




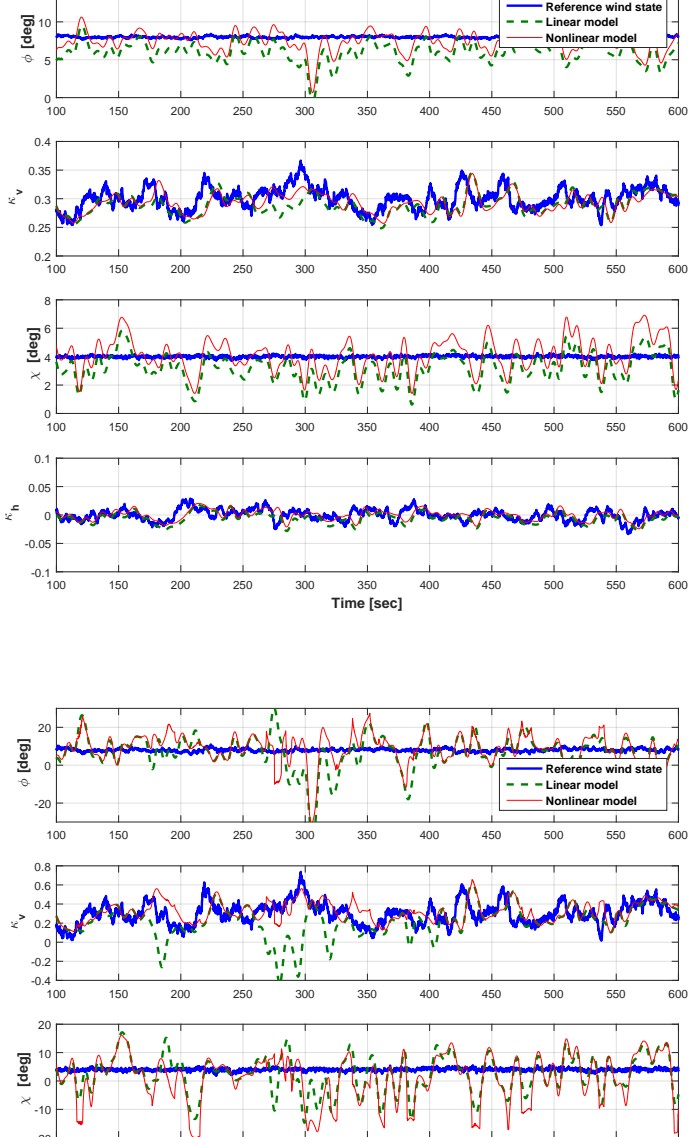

**Figure 13.** Wind state observations in turbulent wind conditions at 19 m/s for a TI equal to 2% (left) and 12% (right). Solid thick blue lines: reference wind parameters; dashed thick green lines: observations by the linear model; solid thin red lines: observations by the nonlinear model.



the shears (see Section 3), errors propagate throughout the solution at a high rate for wind misalignment and upflow, in turn generating fast fluctuations of the estimates.

It should also be remarked that an additional source of uncertainty is the ground truth. In fact, the presence of turbulent eddies in the flow implies that the wind field cannot be exactly parameterized by the assumed wind states. Hence, the reference
quantities plotted here should also be considered as only indicative proxies of the actual wind states.

The observation errors were further analyzed from a statistical standpoint, by generating 5 different turbulent wind field realizations, and computing means and standard deviations. To eliminate the effects of the delay caused by the filter, which would have prevented any instantaneous comparison between the reference and observed quantities, reference wind states were processed with the same filter used for the moment harmonics.

Figure 14 shows the behavior of the standard deviation of the estimation error for the four wind states, as functions of the wind speed and for different TI levels. The curves labeled "TI=0%" refer to the non-turbulent fully-parameterized conditions already described in §4.1. Since a similar behavior characterizes the results of both observers, only those obtained with the nonlinear model are shown here in order not to clutter the figure.

As expected, the standard deviation increases with TI level. Moreover, in regions II and $\mathrm{II}\frac{1}{2}$, accuracy tends to increase for
increasing wind speed, as similarly predicted by the a priori observability analysis. The opposite happens in region III, where oscillations in the results are more significant and strongly affect the estimates. This behavior is particularly visible in the estimation of the angles, as shown in the left plots of Fig. 14

On the other hand, shear errors remain low also at very high TI levels, as illustrated by the right plots of Fig. 14, indicating that fast good quality shear estimates are possible. In fact, for example, the standard deviation of $\kappa_v$ at 7 m/s and 20% TI is
circa 0.055, which means that about 95% of the observer samples have an instantaneous error lower than 0.11.

The evaluation of the observer performance for the angles deserves a special attention. Looking at the yaw misalignment in Fig. 14 (top right) –for regions II, $\mathrm{II}\frac{1}{2}$ and the low region III up to 15 m/s–, the instantaneous error remains within acceptable bounds for turbulence intensities lower than 5%. In fact, $\sigma_\phi$ is lower than 1.5 deg, which implies that estimates are affected by an error lower than 3 deg 95% of the time. On the contrary, the estimation error standard deviation may reach 3, 4 or even
6 deg for the higher turbulence intensities of 12%, 16% and 20%. The maximum error deviation is obtained at 19 m/s for a TI of 12%. The same considerations can be derived for the estimation of the upflow angle.

Figure 15 reports the mean observation errors with respect to the wind speed for both the linear and nonlinear observers. Not unexpectedly, the estimation of the shears is characterized by almost negligible error means. More surprisingly, however, even the mean errors of the angles are quite low for all conditions, although a mild reduction of accuracy can be observed for
increasing wind speeds. In addition, as previously noted, the nonlinear observer appears to be slightly more accurate than the linear one. Finally, it was found that the error means are not significantly influenced by TI.

### 4.2.1 Evaluation of life-time performance

The previous examples have shown that observed angles are typically affected by spurious oscillations, for the reasons explained by the a priori analysis. The same examples however have also shown that mean values are typically well captured,





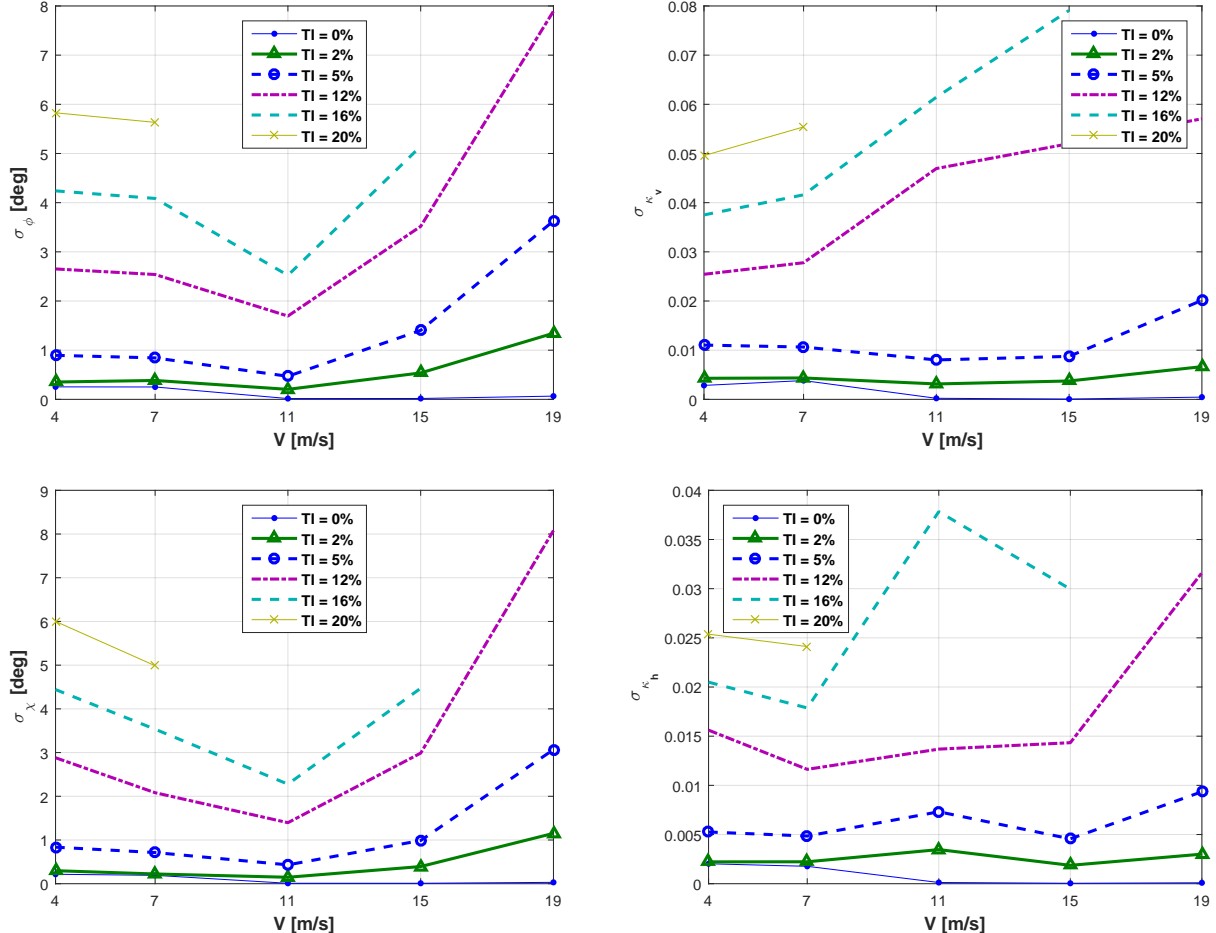

**Figure 14.** Standard deviation of the estimation error of yaw misalingment (top left), vertical shear (top right), upflow angle (bottom left), horizontal shear (bottom right) as functions of wind speed, for varying TI levels. All curves refer to the nonlinear observer results.

and that the amplitude of oscillations is related to TI. This seems to indicate that fast accurate observations of angles are in general not possible, while observations on longer time windows might still be relatively accurate. By the simple inspection of temporal responses, it is however not easy to get a clear idea of the actual precision of the observers in turbulent conditions. In order to provide for a more meaningful indication of the observer accuracy, the "life-time" standard deviation of the observed

5   states is evaluated in this section. This is computed by weighting the results at each wind speed and TI with the corresponding probability distributions at a given site.

To this end, measurements taken at the off-shore platform FINO 1 (FINO) from September 2003 to August 2007 were considered. Figure 16 shows some statistical metrics of the wind at an altitude over the water line of 80 m, which corresponds to the hub-height of the wind turbine considered in the present study. The TI percentiles at 90 m were extracted from Fig. 2 of




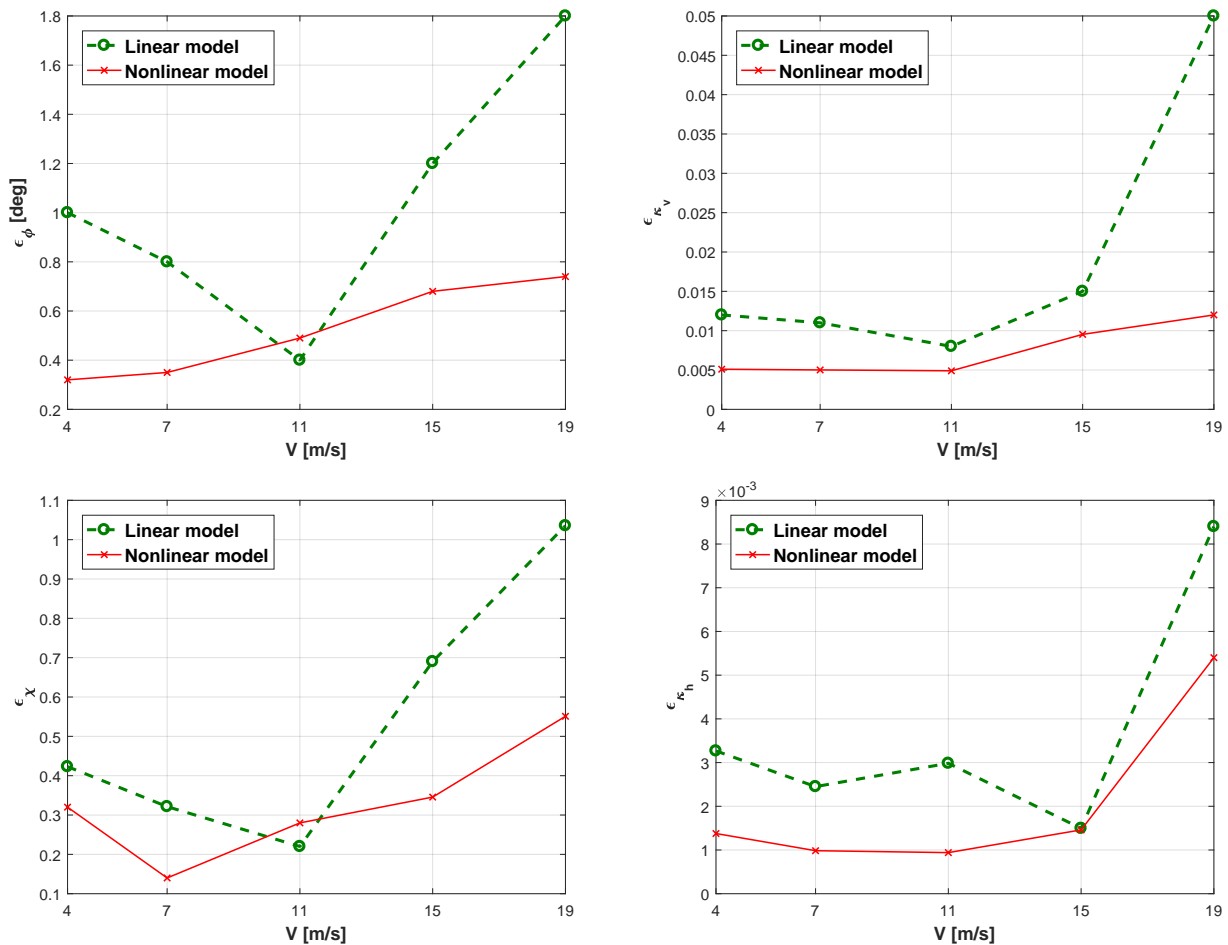

**Figure 15.** Mean estimation error of yaw misalignment (top left), vertical shear (top right), upflow angle (bottom left) and horizontal shear (bottom right) with respect to wind speed.

Türk and Eimeis (2010), and mapped to the current hub-height by scaling with a factor equal to 1.028, according to Fig. 5.21 of Emeis (2013).

Next, a shifted Weibull probability density function (PDF) $\mathcal{W}_\tau$ was fitted to the TI for each wind speed. The PDF is defined as

$$5 \quad \mathcal{W}_\tau(\tau, V) = \begin{cases} \frac{\alpha(V)}{\beta(V)} \left( \frac{(\tau - \tau_{\min}(V))}{\beta(V)} \right)^{\alpha(V)-1} \mathrm{e}^{-\left( \frac{(\tau - \tau_{\min}(V))}{\beta(V)} \right)^{\alpha(V)}}, & \tau \geq \tau_{\min}(V), \\ 0, & \tau < \tau_{\min}(V), \end{cases} \quad (38)$$

while its associated cumulative distribution function (CDF) writes

$$\mathfrak{W}_\tau(\tau, V) = \begin{cases} 1 - \mathrm{e}^{-\left( \frac{(\tau - \tau_{\min})}{\beta(V)} \right)^{\alpha(V)}}, & \tau \geq \tau_{\min}(V), \\ 0, & \tau < \tau_{\min}(V), \end{cases} \quad (39)$$




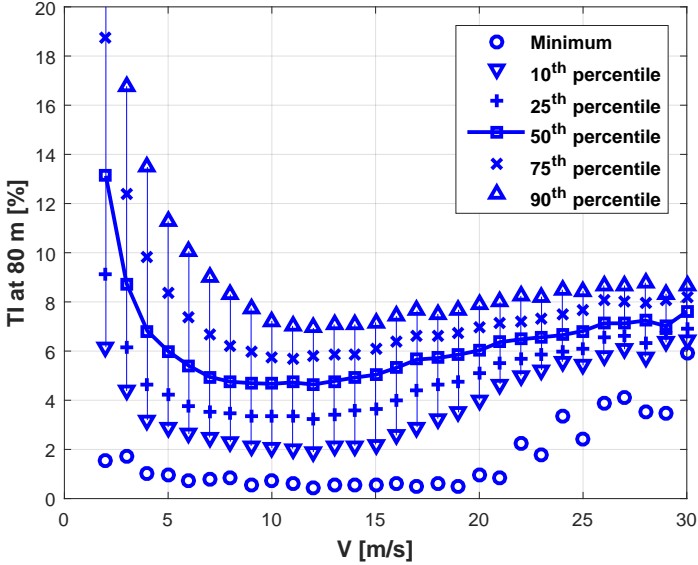

**Figure 16.** Minimum value, $10^{\text{th}}$, $25^{\text{th}}$, $50^{\text{th}}$, $75^{\text{th}}$ and $90^{\text{th}}$ TI percentiles as functions of wind speed, at 80 m above the water line at FINO1 from September 2003 to August 2007. Data taken from Türk and Eimeis (2010).

where $\tau$ is the TI level and $\tau_{\min}(V)$ its minimum value, while $\alpha(V)$ and $\beta(V)$ are the shape and scale parameters, respectively, of the probability density function. Figure 17 represents the Weibull PDF and CDF at 9 m/s.

Given the probability density function of the observation error $\mathcal{P}_\epsilon$, the TI PDF $\mathcal{W}_\tau$ and the wind speed PDF $\mathcal{W}_V$, the life-time standard deviation $\sigma_{\text{LT}}$ can be readily computed as

$$5 \quad \sigma_{\text{LT}} = \frac{1}{\int_{V_{\text{CI}}}^{V_{\text{CO}}} \mathcal{W}_V(V)\mathrm{d}V} \int_{V_{\text{CI}}}^{V_{\text{CO}}} \mathcal{W}_V(V)\left(\int_0^{+\infty} \mathcal{W}_\tau(V,\tau)\left(\int_{-\infty}^{+\infty} \epsilon \mathcal{P}_\epsilon(V,\tau,\epsilon)\mathrm{d}\epsilon\right)\mathrm{d}\tau\right)\mathrm{d}V, \qquad (40)$$

where the innermost integral represents the wind-speed-specific and TI-specific standard deviation of the observation error, $\sigma(V,\tau)$, which was previously computed and reported in Fig. 14. This quantity is then weighted by the probability of each wind speed and TI values to occur at this specific site, as given in Fig. 16.

Figure 18 shows the wind-speed-specific standard deviations for the yaw misalignment and upflow errors, on the left, and

10 for the shear errors, on the right, as well as the wind Weibull distribution at FINO1 as functions of wind speed. The picture clearly illustrates the fact that both for angles and shears, errors are quite limited for the more probable wind speeds.

Finally, the life-time standard deviations are reported in Table 2. From this point of view, results are clearly quite satisfactory not only for shears, but also for angles. In fact, although fluctuations pollute the instantaneous observation of these quantities, their long term metrics are well captured.





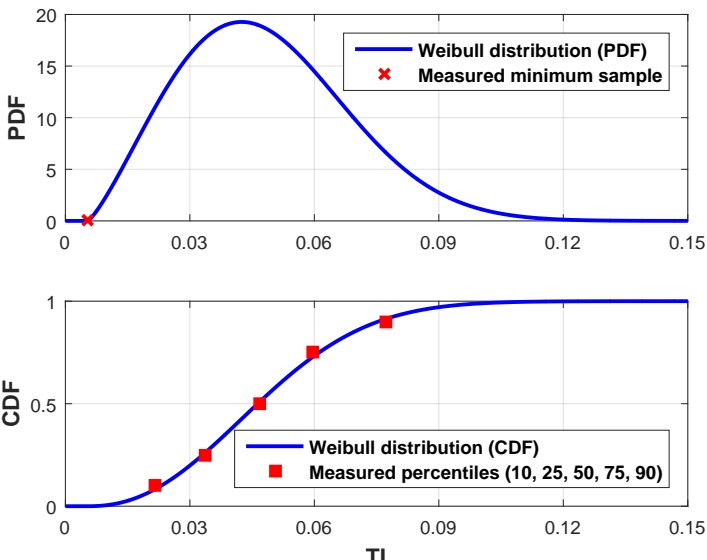

**Figure 17.** TI PDF (upper plot) and CDF (lower plot) at 9 m/s.

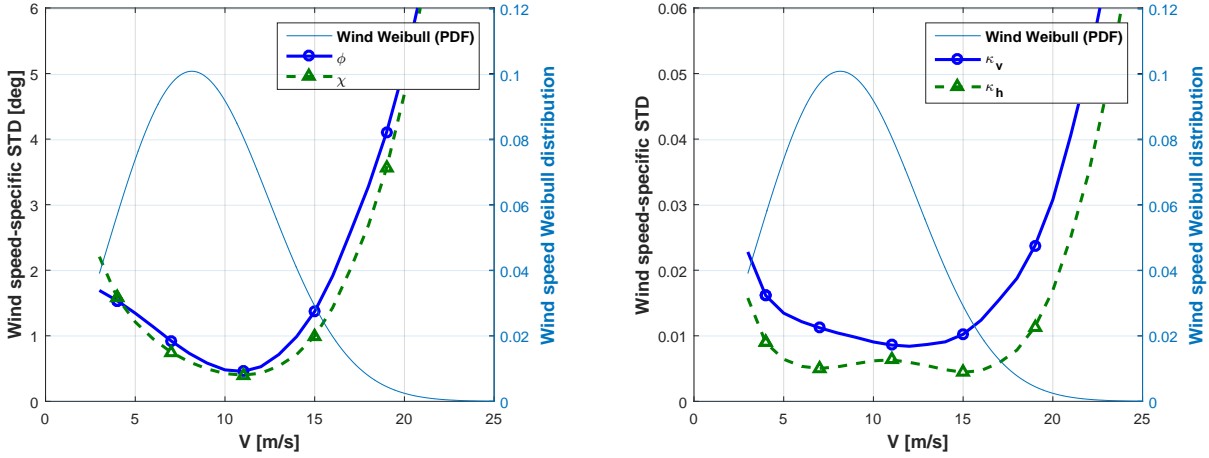

**Figure 18.** Wind-speed-specific standard deviation of the observation error for angles $\phi$ and $\chi$ (left) and for shears $\kappa_v$ and $\kappa_h$ (right). The wind Weibull distribution is characterized by shape and scale parameters equal to 2.5 and 10, respectively.





**Table 2.** Life-time standard deviation and 2-$\sigma$ bounds of the estimation error for the wind parameters.

| Wind parameter | $\phi$ [deg] | $\kappa_v$ | $\chi$ [deg] | $\kappa_h$ |
|---|---|---|---|---|
| $\sigma_{\mathrm{LT}}$ | 0.97 | 0.011 | 0.84 | 0.006 |
| $2\sigma_{\mathrm{LT}}$ | 1.94 | 0.022 | 1.68 | 0.012 |

#### 4.2.2 Following mean changes in yaw misalignment

The fact that the mean estimation errors of the angles, especially for yaw misalignment, are limited, suggests the use of a moving average in order to lower the error standard deviation. This way one may capture the slower variations of the means, while filtering out the faster oscillations. The resulting estimates can be used for slower control actions, as for example yaw
control, or for the slow scale monitoring of parameters of interest.

To test whether it is indeed possible to follow changes of the mean, large changes in yaw misalignment were simulated. Turbulent wind fields were generated with `TurbSim`, and gradually rotated to generate mean wind direction changes from -4 to 4 deg in about 20 seconds. The observed yaw misalignment was filtered with a moving average of variable window length, on account of the mean wind speed. The results of the observations at 7 m/s for different turbulence levels with and without
moving average are shown in Fig. 19.

For the very low TI levels shown in the left plot of Fig. 14, both the mean and instantaneous values of yaw misalignment can be sufficiently well captured even without the use of a filter. On the other hand, with increasing turbulence, spurious oscillations of the estimates mask the mean wind direction change. However, it appears that the use of a moving average is capable of eliminating the faster fluctuations, revealing the presence of a change in wind direction. Clearly, higher values of turbulence
require longer filtering windows, with consequently longer time delays. This delayed detection is however compatible with the usually rather slow and conservative approach used for yaw control, where the actual realignment of the machine is performed only when a wind direction change of some significant entity has been observed for a sufficiently long window of time, usually of many tens of seconds.

As a final remark, the nonlinear observer appears to perform slightly better than the linear one, as more easily visible for low
turbulence conditions.

## 5 Conclusions

This paper has presented a method to estimate the wind inflow at the rotor disk of an operating wind turbine. The proposed method uses the low frequency response of the wind turbine, limited to the 1×Rev harmonics, to infer four wind states representing two misalignment angles and two shears. The rotor response is measured by load sensors, which are becoming




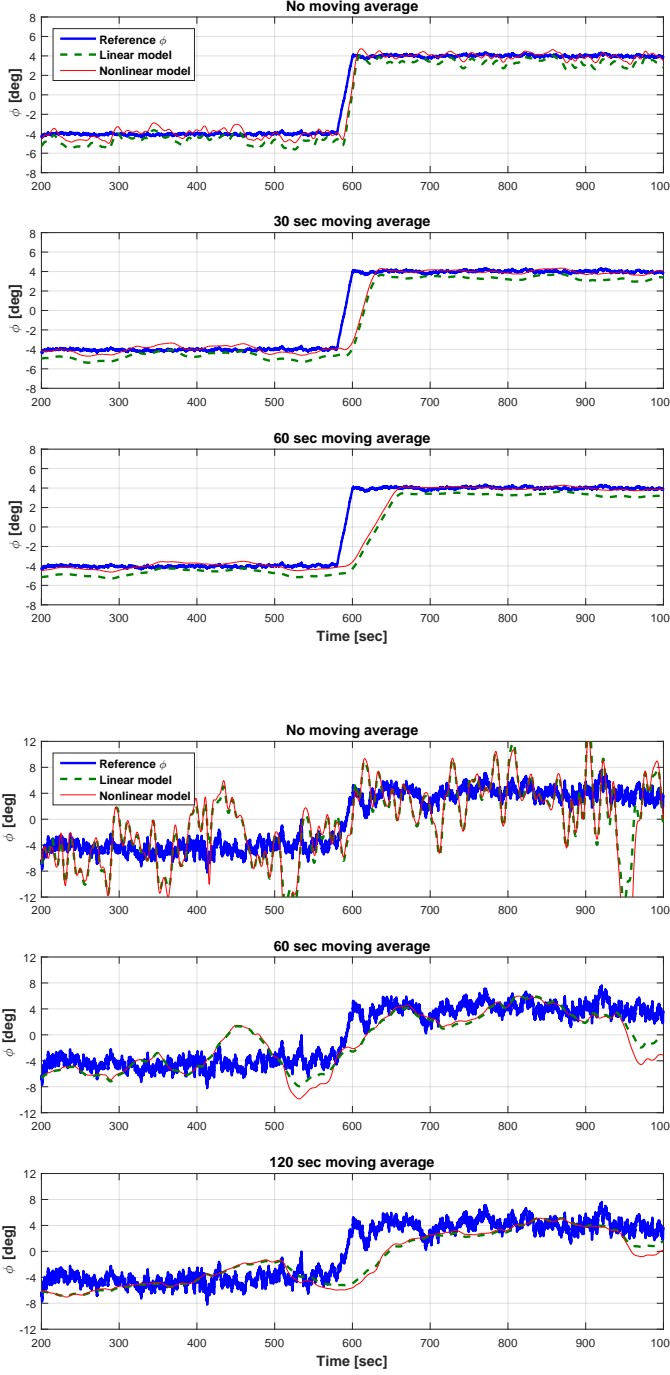

**Figure 19.** Yaw misalignment for an 8 deg change in wind direction at 7 m/s with 2% (left) and 20% (right) TI. Solid thick blue lines: real yaw misalignment; dashed thick green lines: estimate by the linear model; solid thin red lines: estimate by the nonlinear model.

(c) Author(s) 2017. CC BY 3.0 License.





standard equipment on many modern wind turbines. When such sensors are available, the proposed method does not require any additional hardware and amounts to a simple software upgrade.

An input-output model was formulated to represent the relationship between wind states and load harmonics. The model was treated as a black box, whose unknown coefficients were estimated by using the simulated response of a wind turbine implemented in a high-fidelity aeroservoelastic model. The input-output relationship was then inverted in a least-squares sense, in order to provide estimates of the wind states when fed with measured load harmonics. The statistical properties of the model and, in turn, the observability of the wind states were analyzed using the SVD. This a priori analysis highlighted the different nature of the problem of estimating shears and angles, the former being characterized by a higher level of observability than the latter. Finally, the proposed observer was analyzed in a wide range of operating conditions in turbulent wind fields of different characteristics.

From the results of the present study, the following considerations can be made:

– The behavior of the blade out-of-plane and in-plane load harmonics at $1\times$Rev are captured well, both in steady and turbulent conditions, by a linear or second-order nonlinear function of the wind states.

– It is not advisable to include in the model harmonics higher than $1\times$Rev. In fact, although $2\times$Rev components are indeed correlated with wind states, they are also strongly affected by turbulence. In addition, if one uses a simulation model for the estimation or synthesis of the load-wind model, it is to be expected that such a model will better capture the $1\times$Rev response than the higher harmonics. Therefore, limiting load inputs to the $1\times$Rev components helps ensure a higher accuracy of the load-wind model and hence of the estimates.

– Wind states can be estimated from $1\times$Rev blade harmonics, as these quantities carry enough informational content for the model to be invertible.

– An a priori observability analysis shows that the accuracy of the shears is in general superior to the one of the angles. This is not because of a limit of the present specific formulation, but it is due to the intrinsic sensitivity of angle of attack changes to wind state changes, which is different for angles and shears.

– Extensive simulations in turbulent conditions have shown that the mean value of the estimation error is in general significantly low for all states. For example, the mean yaw error is of about 0.5 deg independently of wind speed and TI, whereas the vertical shear error is about 0.01.

– Standard deviations of the shears are in general very low even for high TI levels, implying that the observer is capable of following fast shear fluctuations with good precision.

– Standard deviations for angles are significantly higher, due to their overall lower observability. In general, angle estimates are polluted by rapid spurious oscillations, due to the amplification of errors through the inverted estimation model. This implies that one cannot in general follow rapid variations of the angles, and only observations on longer time scales are possible.



- Although polluted by fluctuations, on average even the angle estimates are of a good quality, thanks also to their small mean errors. An analysis, conducted by taking into account the probability distributions of both wind speed and TI at the off-shore FINO1 platform in the German Bight, has shown that the expected average error in the angles is below 1 deg, which appears to be a very interesting result.

- It was shown that, by filtering the estimated yaw misalignment with a moving average, one may track with good accuracy significant mean changes in the wind direction, indicating the possible use of this estimate for driving the wind turbine yaw control system.

The proposed formulation should be extended to consider the possible presence of an individual pitch control (IPC) strategy. This can be done by including in the load-wind model also the presence of a term depending on pitch load harmonics. As these quantities are known, they represent further inputs that do not change the overall approach, although the model will have additional coefficients that need to be identified. This extension of the formulation has already been tested, and it will be described in a forthcoming publication.

**Nomenclature**

| | |
|---|---|
| $m$ | Generic blade moment |
| $t$ | Time |
| $n_k$ | Piecewise linear shape function |
| $q$ | Dynamic pressure |
| $p$ | Order of the nonlinear model |
| $f$ | Frequency |
| | |
| $B$ | Number of blades |
| $V$ | Wind speed |
| $R$ | Rotor radius |
| $H$ | Hub height |
| $\mathcal{V}$ | Variance |
| $N_{\mathrm{obs}}$ | Number of experiments for model identification |
| $N_{\mathrm{nodes}}$ | Number of nodes for wind speed scheduling |
| $\mathfrak{R}$ | Real number set |
| $\mathcal{W}$ | Weibull probability density function |
| $\mathfrak{W}$ | Weibull cumulative distribution function |
| $\mathcal{P}_{\epsilon}$ | Probability density function of the observation error |
| $V_{\mathrm{CI}}$ | Cut-in wind speed |



| | | |
|---|---|---|
| $V_{\mathrm{CO}}$ | | Cut-out wind speed |
| $V_{\mathrm{RP}}$ | | Rated wind speed |
| | | |
| $\boldsymbol{h}$ | | Vector of harmonic amplitudes for signal demodulation |
| 5 | $\boldsymbol{y}$ | Output vector |
| | $\boldsymbol{x}$ | State vector |
| | $\boldsymbol{u}$ | Input vector |
| | $\boldsymbol{m}$ | Vector of moment harmonics |
| | $\boldsymbol{r}$ | Measurement error |
| 10 | | |
| | $\boldsymbol{X}$ | Demodulation matrix |
| | $\boldsymbol{1}$ | Unitary vector |
| | $\boldsymbol{R}$ | Measurement error covariance matrix |
| | $\boldsymbol{U}$ | Matrix of left singular vectors |
| 15 | $\boldsymbol{V}$ | Matrix of right singular vectors |
| | $\boldsymbol{I}$ | Identity matrix |
| | $\mathcal{M}$ | Steady-state relation between load harmonics and wind state vector |
| | | |
| | $\boldsymbol{E}[\,\cdot\,]$ | Expected value |
| 20 | $\mathrm{Cov}[\,\cdot\,]$ | Covariance |
| | | |
| | $\varrho$ | Air density |
| | $\phi$ | Yaw misalignment angle |
| | $\chi$ | Upflow angle |
| 25 | $\kappa_v$ | Vertical shear |
| | $\kappa_h$ | Horizontal shear |
| | $\psi$ | Azimuth angle |
| | $\sigma$ | Standard deviation |
| | $\alpha$ | Shape parameter of the Weibull distribution |
| 30 | $\beta$ | Scale parameter of the Weibull distribution |
| | $\tau$ | Turbulence intensity level |
| | $\zeta$ | Blade section inflow angle |
| | $\Omega$ | Rotor angular velocity |
| | | |
| 35 | $\epsilon_{\boldsymbol{\theta}}$ | Wind state observation error |





| | | |
|---|---|---|
| $\boldsymbol{\xi}$ | | Vector of statistically independent wind state variables, $\boldsymbol{V}^T\boldsymbol{\theta}$ |
| $\boldsymbol{\theta}$ | | Wind state vector |
| $\boldsymbol{\Sigma}$ | | Rectangular matrix of singular values |
| | | |
| 5 | $(\cdot)^T$ | Transpose |
| | $(\cdot)^{(i)}$ | Quantity related to the $i$th experiment |
| | $(\cdot)_{(j)}$ | Quantity related to the $j$th blade |
| | $(\cdot)_k$ | Nodal quantity at the $k$th node |
| | $(\cdot)^{\mathrm{OP}}$ | Out-of-plane quantity |
| 10 | $(\cdot)^{\mathrm{IP}}$ | In-plane quantity |
| | $\dot{(\cdot)}$ | Time derivative, i.e. $\mathrm{d}\cdot/\mathrm{d}t$ |
| | $(\cdot)_{n\mathrm{c}}$ | $n\times$Rev cosine amplitude |
| | $(\cdot)_{n\mathrm{s}}$ | $n\times$Rev sine amplitude |
| | $(\cdot)_{\mathrm{M}}$ | Measured quantity |
| 15 | $(\cdot)_{\mathrm{E}}$ | Estimated quantity |
| | $(\cdot)_{\mathrm{R}}$ | Real quantity |
| | $(\cdot)_{\mathrm{NL}}$ | Nonlinear term |
| | | |
| | BEM | Blade element momentum |
| 20 | SVD | Singular Value Decomposition |
| | LiDAR | Light detection and ranging |
| | TSR | Tip speed ratio |
| | PDF | Probability density function |
| | CDF | Cumulative distribution function |
| 25 | IPC | Individual pitch control |
| | TI | Turbulence intensity |
| | STD | Standard deviation |





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
