# Peer review of "Wind inflow observation from load harmonics"

_Wind Energy Science, 2017_

## Referee Comment (RC1) · Anonymous Referee #1 · 26 Jul 2017

Excellent paper with clear explanations and detailed analysis. The introduction and conclusion sections are especially well crafted in explaining the motivation, background references and motivations for the work. The analysis is convincing and thorough. Great!

Some comments:

One detail I struggled to comprehend was the distinctions made on page 8 and continuing, regarding white box (I don't know what this is, could you define?), gray box and black box. The authors say they are switching to black-box from page 8 forward, and yet it still seems a lot of internal structure is maintained, for example which harmonics to include in the model. Somehow when I hear black-box, I think of completely blind structures, like neural networks, where there is an input-output relationship, but the internal structure is not physically meaningful in anyway. I prefer the approach in

the paper which I would have thought is gray in that it seems like estimation of harmonic coefficients, but is that a misunderstanding? Or is the translation to harmonics performed external to the black box?

Also on page 8 it is said that the disadvantage of black-box is a necessary rich data-set. I typically add that the internals of a black-box are perhaps without physical interpretation, but this is somehow not true of this work? Sorry for a long point, but perhaps some further explanation around this point could be included.

Another point of confusion for me is at the top of page 14, there is a 2xRev harmonic discussion, but I thought I had understood from earlier explanations that only 1xRev harmonics are included because of challenges in interpreting 2XRev.

Small comments: equation (37), what is meant by the tildas? figure 15, what is the averaging window length?

---

## Referee Comment (RC2) · Anonymous Referee #2 · 2 Aug 2017

A linear and non-linear formulations of a black box model, which relates wind turbine loads and wind state inflows (shears and misalignments), were formulated. Coefficients of this model were identified using a dataset from an aeroelastic model. First results of the identification were validated using different wind state inflows (steady, turbulent). It was found that 2-Rev harmonics of loads are not well reproduced by the model (even when using the non-linear one), while the 1-Rev harmonics is very well reproduced. The system was then inverted to infer wind states from wind turbine loads. From this formulation, an observability analysis was performed which main conclusion is that shears contributes in majority to out-of-plane bending loads while misalignments are more related to in-plane bending loads. Also shears are much more observable than misalignments angles. This is attributed latter to a greater sensitivity to changes in shear on the inflow angle (and thus lift or loads) than to changes in wind state angles. The inverted system was finally tested on a dataset from an aeroelastic model

using non-turbulent or turbulent (Kaimal distribution with different turbulent intensities from TURBSIM software) wind state inflows. Note that authors uses an instantaneous parametrization of the turbulence using wind state equations described in the paper. Results show that the shears are better inferred than misalignments as previously expected. Also, an increase of the turbulent intensity increases the standard deviation of the estimation error, mainly for wind state angles. From an observability analysis of the model using a much longer inflow dataset (4 years), the estimation of the wind state angles is expected to be improved. Also, simulations were performed with a progressive yaw misalignment to evaluate the model to infer the wind state angle changes. It is found that, with high turbulence intensity, applying a moving averaged filter on the inferred wind state angles delay the response but predict well the level of changes.

This is an interesting subject and a consequent work on ways of using the wind turbine observers (here load sensors) to predict wind inflow states. It follows articles from the same authors: [1] C.L.Bottasso and C.E.D.Riboldi (2014) "Estimation of wind misalignment and vertical shear from blade loads", Renewable Energy, 62:293-302, https://doi.org/10.1016/j.renene.2013.07.021 [2] C.L.Bottasso and C.E.D.Riboldi (2015) "Validation of a wind misalignment observer using field test data", Renewable Energy, 74:298-306, https://doi.org/10.1016/j.renene.2014.07.048

Compared to reference [1] the present study uses a black box model and 4 degrees of wind state inflows (2 shears and 2 misalignment angles). Also a non-linear formulation of the model is tested. It has to be emphasis that this study is based on simulations from an engineering software (with all associated simplifications) and uses 2500 degrees of freedom as potential load sensors. Maybe an interesting future work would be to evaluate the minimal load sensors required for an acceptable infer of the wind inflow states.

I think the additional work performed in the present study compared to previous work (reference 1) is significant and interesting, so that the article worth being published in the Wind Energy Science journal with however some corrections listed below (ordered

as appearing in the document):

P 2 L 10: "The problem of mapping the information from a met-mast to the rotor disk of a wind turbine is in general very difficult to solve, and it will clearly be always prone to possibly severe inaccuracies."

What make the inverse, sensing the wind from the rotor response, easier ?

One reason evocate by the authors:

- rotor response will be non-local and rotor-effective → but the global rotor response is reconstructed from local sensors at the blade which may encounter local phenomena such as the flow locally separated ...

P3 L13 "motivated by the very promising results in the field ..." Certainly a confusion with the work of the same authors in 2015 ?

p3 L22: authors uses the simplified notation 1-Rev and 2-Rev that is not introduced, please be more explicit. Such as "first rotor revolution" . . .

P15 L18 "V=(u,v,z)T" Replace z by w.

p7 L22: Can you please give the relation between y and m ?

p8 L30-33: " State-of-the-art aeroservoelastic codes used for the design and certification of wind turbines . . . with experimental data" Can you please give some references of this type of comparison. In particular it is interesting to know the limitations.

P9L25: " Separating the effects of gravity . . . affecting load measurements." ? Figure 2 do not exhibit a significant sensitivity to rho in contradiction to what is said here. Please explain.

P11 L12: region II and III are not explicitly written in figure 2, where are they ? I guess they are delimited by red lines ?

P12 L14: "A rather wide transition ..." Does this description correspond to figure 2 ?

p12L27: section 2.3.5 describes an example of the linear/non linear models (Eq. 11A and 22) identification using a dataset from the aeroelastic simulation model.

However, it is not indicated where loads m are taken to identify the model ? Unless all degree of freedoms of the code are used ? If so, it would be interesting to know the minimal set of data required and where they should be placed for an accurate identification.

P15 L11: "The reason for the very poor results of the turbulent case ... this information cannot be separated from the pollution brought by smaller scale wind field fluctuation."

Does it mean that turbulent small scales do not contribute significantly to the load 2-Rev harmonic in the aeroelastic model ? Or maybe the turbulent fluctuation can't be described by the wind state equations given by the authors ?

P20: It is not clear if results of the observability analysis are presented from the linear or non-linear model formulation. It seems that results 37 are only expressed from the linear formulation. Is it correct ?

P20L9-10: "In the third case, the noise covariance ... and the ones measured on the simulation model, msim, ie " It is not clear here what wind state inputs are taken to get mobs and msim.

P20L24: "On the other hand, ... do not indicate a predominant effect on some load components" From results indicated in the paper, I see a significant dominance of In-plane loads (up to 3 times higher than out-of-plane loads). This is not "a bit higher" as indicated later.

P20L27: "A side observation, ... in the response of the machine" 1- I see a line inversion (cosinus components of loads becomes sinus components of loads from one wind state couple to the other wind state couple) but no obvious type of symmetry. Please be more explicit.

2- Also, this 90-symmetry in the wind states indicate that one couple can be determined

by the other couple, so that the problem may be reduced ? If so, this is in contradiction with the introduction p3L15 "First extensive numerical experiments have shown that the load-wind model on which the estimator is based must consider at least four wind states instead of two"

3- What about the non-linear model ? I guess we expect less symmetry and then a higher problem to solve, but what about the distribution of the observability on load components for the different wind states ?

p22L2: "Given the behavior of the linear and nonlinear observers" From what I read of the paper, only results from the linear observers are given, I'm right ? Please be clearer in the observer type you are using in section 3.1.

p24 figure 11: It is impossible to see the legend and plots are too small. Please make it larger.

P26 figure 12 and 13: "left" and "right" should be replaced by "up" and "down".

P25: For this part (section 4.2) are you using all degree of freedoms (2500 blade loads) ? It is not clear in the paper as you are talking about "wind conditions at the location occupied by each single blade". Please make it clearer.

P28L31: "Finally, it was found that the error means are not significantly influenced by TI" This is not shown in figure 15.

p28 section 4.2.1: I guess you don't have any load measurements, so you use the model identified previously from the aeroelastic simulations ? This means that you evaluate the observability from the off-shore platform dataset assuming the identified model is identical as the wind turbine is identical, i'm right ? Please make it clearer.

p30L1: Please report figure 5.21 of Emeis(2013) in the article, it is just annoying to have to find (buy) a book to follow what the authors wants to say.

P31 figure 16: from the article of Türk and Eimeis (2010), TI is reported to be measured

at 90m not 80m.

P34 figure 19: change left/right to up/down

p36L3: "the expected average error in angles is below 1deg" This is the expected mean estimation error based on observability formulations, you are not able to compare with real measurements, I'm wrong ? This also suppose you measure loads from 2500 sensors on the blades, I'm wrong ?

---

## Author Comment (AC1) · 31 Aug 2017

**Reply to Reviewer 1**

We thank Reviewer 1 for the detailed analysis and constructive inputs. A list of point-by-point replies to the Reviewer's comments is detailed in the following.

**Reviewer** *Excellent paper with clear explanations and detailed analysis. The introduction and conclusion sections are especially well crafted in explaining the motivation, background references and motivations for the work. The analysis is convincing and thorough. Great!*

**Authors** Thanks!

**Reviewer** *One detail I struggled to comprehend was the distinctions made on page 8 and continuing, regarding white box (I don't know what this is, could you define?), gray box and black box. The authors say they are switching to black-box from page 8 forward, and yet it still seems a lot of internal structure is maintained, for example which harmonics to include in the model. Somehow when I hear black-box, I think of completely blind structures, like neural networks, where there is an input-output relationship, but the internal structure is not physically meaningful in anyway. I prefer the approach in the paper which I would have thought is gray in that it seems like estimation of harmonic coefficients, but is that a misunderstanding? Or is the translation to harmonics performed external to the black box? Also on page 8 it is said that the disadvantage of black-box is a necessary rich data-set. I typically add that the internals of a black-box are perhaps without physical interpretation, but this is somehow not true of this work? Sorry for a long point, but perhaps some further explanation around this point could be included.*

**Author** We have added the definitions of white, gray and black box models, and we also cited the following reference on this topic: Ljung, L.: Perspectives on system identification, Annual Reviews in Control, 34(1), 1–12, 2010.

The reviewer is right in pointing out that, while on the one hand we use a black box (blind) approach, on the other we bring some structure (and knowledge) into it when we defined as outputs the load harmonics. However, this operation comes prior to the definition of the model that, as a result, is defined as in Eq. (10). We have modified the text to make this point clearer.

**Reviewer** *Another point of confusion for me is at the top of page 14, there is a 2xRev harmonic discussion, but I thought I had understood from earlier explanations that only 1xRev harmonics are included because of challenges in interpreting 2XRev.*

**Authors** The fact that only $1\times$Rev harmonics should be included in the models is one

of the results of the paper, and it is anticipated in the Introduction. At page 14 this results has not yet been obtained, and the paper still treats the general case of multiple harmonics. It is only immediately later on, in section 2.3.6, that the reason why $2\times$Rev should not be considered is explained.

**Reviewer** *Small comments: equation (37), what is meant by the tildas?*

**Authors** The symbol means 'approximately'. We added the definition of this notation immediately after Eq. (37).

**Reviewer** *figure 15, what is the averaging window length?*

**Authors** 10 minutes, and this has now been added to the text.

We have taken the opportunity to make several small editorial changes to the text, in order to improve readability. A revised version of the manuscript is attached to the present reply, with the main changes highlighted in red.

Best regards.
The authors

[Figure]

**Supplement:**

[revised manuscript text omitted]
}[\boldsymbol{\epsilon}_\theta] = \mathcal{E}[\boldsymbol{\epsilon}_\theta \boldsymbol{\epsilon}_\theta^T]$ (Cramer, 1946) writes

$$\mathrm{Cov}[\boldsymbol{\epsilon}_\theta] = \left(\boldsymbol{F}^T \boldsymbol{R}^{-1} \boldsymbol{F}\right)^{-1}. \tag{32}$$

This expression shows the interplay between noise $\boldsymbol{r}$ and sensitivity $\boldsymbol{F}$, captured by the term $\boldsymbol{R}^{-\frac{1}{2}}\boldsymbol{F}$: the higher the variance and/or the lower the sensitivity of the measurements with respect to the wind states, the worst the accuracy of the estimates.

The covariance $\mathrm{Cov}[\boldsymbol{\epsilon}_\theta]$ expressed by Eq. (32) is typically fully populated, as the errors of the estimates are correlated. To ease the understanding of the estimation problem, the SVD (Golub and van Loan, 1996) can be used to decouple the estimates. In fact, matrix $\boldsymbol{R}^{-\frac{1}{2}}\boldsymbol{F}$ can be factored as

$$\boldsymbol{R}^{-\frac{1}{2}}\boldsymbol{F} = \boldsymbol{U}\boldsymbol{\Sigma}\boldsymbol{V}^T, \tag{33}$$

where $\boldsymbol{U} \in \Re^{m \times m}$, $\boldsymbol{\Sigma} \in \Re^{m \times n}$ and $\boldsymbol{V} \in \Re^{n \times n}$, being $m$ the number of measurements and $n$ the number of wind state variables. Matrices $\boldsymbol{U}$ and $\boldsymbol{V}$ are orthonormal, i.e. $\boldsymbol{U}^T\boldsymbol{U} = \boldsymbol{U}\boldsymbol{U}^T = \boldsymbol{I}$ and $\boldsymbol{V}^T\boldsymbol{V} = \boldsymbol{V}\boldsymbol{V}^T = \boldsymbol{I}$, whereas $\boldsymbol{\Sigma} = \mathrm{diag}(\dots, 1/\sigma_i, \dots)$ is a diagonal matrix and $\sigma_i$ the standard deviation. Inserting Eq. (33) into Eq. (32), the covariance of the estimation error can be expressed as

$$\mathrm{Cov}\left[\boldsymbol{V}^T\boldsymbol{\epsilon}_\theta\right] = \boldsymbol{E}\left[\left(\boldsymbol{V}^T(\boldsymbol{\theta}_E - \boldsymbol{\theta}_R)\right)\left(\boldsymbol{V}^T(\boldsymbol{\theta}_E - \boldsymbol{\theta}_R)\right)^T\right] = \left(\boldsymbol{\Sigma}^T\boldsymbol{\Sigma}\right)^{-1} = \mathrm{diag}(\dots, \sigma_i^2, \dots). \tag{34}$$

This way, the problem is reformulated by the change of variables $\boldsymbol{\xi} = \boldsymbol{V}^T\boldsymbol{\theta}$, where $\boldsymbol{\xi}$ are statistically independent variables with diagonal covariance. This reformulation simplifies the interpretation of the structural observability of the problem. In fact, the $i$th column of matrix $\boldsymbol{V}$ linearly combines the wind parameters, mapping them into a new parameter $\xi_i$ with variance $\sigma_i^2$. Clearly, a high variance indicates a low level of identifiability of the associated linear combination of wind parameters.

This analysis also provides information on the dependence of loads on wind states. In fact, one can easily show that

$$\frac{\partial \boldsymbol{m}}{\partial \boldsymbol{\xi}} = \boldsymbol{R}^{\frac{1}{2}}\boldsymbol{U}\boldsymbol{\Sigma}. \tag{35}$$

Therefore, the analysis of $\boldsymbol{U}$ reveals on which linear combination of inflow parameters each load depends the most.

The same analysis can be applied to the nonlinear case, by linearizing Eq. (22) around a specific operating and wind condition and using $\boldsymbol{F} = \partial(\boldsymbol{F}_{\mathrm{NL}}\boldsymbol{\theta}_{\mathrm{NL}})/\partial\boldsymbol{\theta} = \boldsymbol{F}_{\mathrm{NL}}\partial\boldsymbol{\theta}_{\mathrm{NL}}/\partial\boldsymbol{\
[revised manuscript text omitted]

---

## Author Comment (AC2) · 31 Aug 2017

**Reply to Reviewer 2**

We thank Reviewer 2 for the detailed analysis and constructive inputs. A list of point-by-point replies to the Reviewer's comments is detailed in the following.

**Reviewer** *P 2 L 10: "The problem of mapping the information from a met-mast to the rotor disk of a wind turbine is in general very difficult to solve, and it will clearly be always prone to possibly severe inaccuracies." What make the inverse, sensing the wind from the rotor response, easier ? One reason evocate by the authors: rotor response will be non-local and rotor-effective! but the global rotor response*

*is reconstructed from local sensors at the blade which may encounter local phenomena such as the flow locally separated . . .*

**Authors** We agree, but the sensors of all blades are used simultaneously, which has the clear effect of averaging the information over the rotor disk. In addition, load harmonics are used, which has the additional effect of implying an azimuthal averaging. Hence local effects are filtered out, and only rotor-effective quantities are obtained.

**Reviewer** *P3 L13 "motivated by the very promising results in the field . . ." Certainly a confusion with the work of the same authors in 2015?*

**Authors** We are unsure about the meaning of this comment. With that sentence we refer to the approach developed in Bottasso and Riboldi (2014), which was partially validated with field tests in Bottasso and Riboldi (2015). Specifically, the results obtained in the field validation showed that the model formulation was promising, although still incomplete as already clearly explained in the introduction: "Notwithstanding these very promising results, the same study also showed a marked sensitivity of the results on the wind upflow angle, indicating the probable need for a richer description of the wind field." Therefore, instead of observing only vertical shear and yaw misalignment as in Bottasso and Riboldi (2014) and (2015), in the present paper we include two new states in the formulation, in order to fully parameterize the wind inflow and improve the observer performance.

**Reviewer** *P3 L22: authors uses the simplified notation 1-Rev and 2-Rev that is not introduced, please be more explicit. Such as "first rotor revolution". . .*

**Authors** The meaning of the notation $n \times$ Rev was added in the introduction.

**Reviewer** *P7 L22: Can you please give the relation between y and m ?*

**Authors** This has now been added to the text.

**Reviewer** *P8 L30-33: "State-of-the-art aeroservoelastic codes used for the design and certification of wind turbines . . . with experimental data". Can you please give some references of this type of comparison. In particular it is interesting to know the limitations.*

**Authors** We are not aware of one single good review on the problem of validation of aeroelastic codes, also probably because of a lot of these activities are done internally by wind turbine manufacturers. In any case, this seems to be beyond the scope of the present paper. To accommodate the request of the reviewer we have eliminated part of the sentence.

**Reviewer** *P9 L25: "Separating the effects of gravity . . . affecting load measurements."? Figure 2 do not exhibit a significant sensitivity to rho in contradiction to what is said here. Please explain.*

**Authors** We do not agree. Figure 2 shows that at 3 m/s a typical derivative decreases of about 20% when moving from $\rho = 1.1$ to $1.35$ kg/m$^3$, while at 11 m/s the same quantity increases by 30%. This is only to be expected: aerodynamic loads do depend on density, which does indeed depend on ambient conditions.

**Reviewer** *P11 L12: region II and III are not explicitly written in figure 2, where are they ? I guess they are delimited by red lines ?*

**Authors** That is correct, and we have now noted this in the caption of the figure. Details about the operating regions of this machine can be found in Section 2.3.4.

**Reviewer** *P12 L14: "A rather wide transition . . . " Does this description correspond to figure 2 ?*

**Authors** Yes. In Fig. 2 the transition region is delimited by red lines, as now explained in the caption.

**Reviewer** *P12 L27: section 2.3.5 describes an example of the linear/non linear models (Eq. 11A and 22) identification using a dataset from the aeroelastic simulation*

model. However, it is not indicated where loads m are taken to identify the model ? Unless all degree of freedoms of the code are used ? If so, it would be interesting to know the minimal set of data required and where they should be placed for an accurate identification.

**Authors** Loads are taken from measurements, for example through strain gages. The beginning of Sect. 2.1 states ". . . the present observer is based on the lowest load harmonics. Although other response indicators could be used in principle, as for example accelerations, loads are considered in this work because they are now often measured on board modern large wind turbines for enabling load-feedback control, and load sensors will probably be standard equipment available on most future machines." In addition, the Introduction explains at length the concept.

We do not understand the comment "Unless all degree of freedoms of the code are used". Clearly the number of degrees of freedom of the model will in general affect its fidelity to reality. However this has nothing to do with the measurement of blade loads in the context of the present paper.

**Reviewer** *P15 L11: "The reason for the very poor results of the turbulent case . . . this information cannot be separated from the pollution brought by smaller scale wind field fluctuation." Does it mean that turbulent small scales do not contribute significantly to the load 2-Rev harmonic in the aeroelastic model? Or maybe the turbulent fluctuation can't be described by the wind state equations given by the authors?*

**Authors** None of the two. It means that both wind states and turbulent fluctuations induce 2-Rev harmonics of comparable magnitude. Hence, the two effects cannot be separated. We believe this is clearly explained in our text: "It should be remarked that, as previously illustrated in Fig. 4, the model is perfectly capable of capturing with good accuracy these higher harmonics in steady wind conditions. The reason for the very poor results of the turbulent case is due to the fact that small scale turbulent fluctuations in the wind field cause $2 \times$Rev harmonics that

are comparable to, if not larger than, the ones caused by the wind states used for the parameterization. Therefore, although $2\times$Rev harmonics carry information on the wind states, this information cannot be separated from the pollution brought by the smaller-scale wind field fluctuations. In this sense, $2\times$Rev harmonics are not good candidates for the observation of wind states."

**Reviewer** *P20: It is not clear if results of the observability analysis are presented from the linear or non-linear model formulation. It seems that results 37 are only expressed from the linear formulation. Is it correct?*

**Authors** The last sentence of Section 3.1 states "Similar results, not shown here for the sake of brevity, were obtained with the nonlinear model.".

**Reviewer** *P20 L9-10: "In the third case, the noise covariance . . . and the ones measured on the simulation model, msim, ie". It is not clear here what wind state inputs are taken to get mobs and msim.*

**Authors** This part was rephrased to improve clarity.

**Reviewer** *P20 L24: "On the other hand, . . . do not indicate a predominant effect on some load components" From results indicated in the paper, I see a significant dominance of Inplane loads (up to 3 times higher than out-of-plane loads). This is not "a bit higher" as indicated later.*

**Authors** We agree, and removed "bit".

**Reviewer** *P20 L27: "A side observation, . . . in the response of the machine"*

1. *I see a line inversion (cosinus components of loads becomes sinus components of loads from one wind state couple to the other wind state couple) but no obvious type of symmetry. Please be more explicit.*
2. *Also, this 90-symmetry in the wind states indicate that one couple can be determined by the other couple, so that the problem may be reduced? If so,*

*this is in contradiction with the introduction p3 L15 "First extensive numerical experiments have shown that the load-wind model on which the estimator is based must consider at least four wind states instead of two"*

3. *What about the non-linear model? I guess we expect less symmetry and then a higher problem to solve, but what about the distribution of the observability on load components for the different wind states?*

**Authors** The text was updated to make this point clearer. In particular:

1. With the term symmetry here we are referring to the fact that the rotor response to a vertical shear is the same as for a horizontal shear, but shifted by 90 deg. The same holds for the angles. That is why we obtained $U_{11} = U_{22}$ and $U_{21} = -U_{12}$. The text was improved to clarify this point.

2. It is true that, given the symmetry in the response, the identification problem can be simplified. However, this is not in contradiction to what is written at page 3 line 15. In fact, the model should consider the entire set of wind states, made by four parameters, to coherently describe the wind field. This can be seen even by the SVD analysis: each wind state is affecting at least one of the 1-Rev components. What could be simplified/improved is only the identification step. This point has now been briefly added to the text. Furthermore, we have cited an additional reference that further explains the use of symmetry in the response of the rotor (Cacciola, Bertelé, Schreiber and Bottasso, Journal of Physics: Conference Series 753(3):032036)).

3. How to exploit the isotropic behavior of the rotor in the case of a nonlinear model is a point that deserves further investigation. However, as explained above, the symmetry in the response can only be used to simplify the identification, but will not drastically change the results. Therefore, the fact that we did not use symmetry here for the linear and nonlinear models does not affect results and conclusions of the present work.

**Reviewer** *P22 L2: "Given the behavior of the linear and nonlinear observers" From what I read of the paper, only results from the linear observers are given, I'm right? Please be clearer in the observer type you are using in section 3.1.*

**Authors** As previously explained, both linear and nonlinear models were considered in the a priori observability analysis, and this is stated in the text.

**Reviewer** *P24 figure 11: It is impossible to see the legend and plots are too small. Please make it larger.*

**Authors** Figure and legend were made bigger.

**Reviewer** *P26 figure 12 and 13: "left" and "right" should be replaced by "up" and "down".*

**Authors** The captions were modified as suggested.

**Reviewer** *P25: For this part (section 4.2) are you using all degree of freedoms (2500 blade loads) ? It is not clear in the paper as you are talking about "wind conditions at the location occupied by each single blade". Please make it clearer.*

**Authors** As stated in the paper, linear and nonlinear observers use as input the $1 \times \text{Rev}$ cosine and sine components of the in- and out-of-plane bending moments at blade root. Therefore we do not understand the reference to the 2500 dofs of the model. As the number of dofs in the model is totally irrelevant, we have eliminated its mention from the text.

**Reviewer** *"Finally, it was found that the error means are not significantly influenced by TI" This is not shown in figure 15.*

**Authors** We changed the sentence to make it clearer.

**Reviewer** *P28 section 4.2.1: I guess you don't have any load measurements, so you use the model identified previously from the aeroelastic simulations? This means that you evaluate the observability from the off-shore platform dataset assuming*

*the identified model is identical as the wind turbine is identical, i'm right? Please make it clearer.*

**Authors** Here we are assessing the precision of our method if it were employed on a machine like the one used in this work but located at the off-shore platform FINO 1. Wind measurements of a period of about 4 years were considered to observe the probability of each TI to occur at each wind speed. With such information, along with the Weibull distribution, we can know which TI characterizes the site and therefore calculate which error deviation to expect. A sentence was added before the explanation of Fig. (16), as suggested by the Reviewer, to clarify this point.

**Reviewer** *P30 L1: Please report figure 5.21 of Emeis(2013) in the article, it is just annoying to have to find (buy) a book to follow what the authors wants to say.*

**Authors** Figure 5.21 of Emeis(2013) shows nothing but the variation of TI as a function of hub height, which is more or less linear at the height of interest. Since it is a really simple plot, instead of inserting another figure (which would also necessitate obtaining the necessary authorization because of copyright), we preferred to add a better description.

**Reviewer** *P31 figure 16: from the article of Türk and Eimeis (2010), TI is reported to be measured at 90m not 80m.*

**Authors** Correct, and this is why the data was scaled down to 80 m with a factor equal to 1.028, as explained in the text.

**Reviewer** *P34 figure 19: change left/right to up/down*

**Authors** This was corrected as suggested.

**Reviewer** *P36 L3: "the expected average error in angles is below 1 deg" This is the expected mean estimation error based on observability formulations, you are not able to compare with real measurements, I'm wrong ? This also suppose you measure loads from 2500 sensors on the blades, I'm wrong ?*

**Authors** This result is not based on real measurements but on the simulated behavior of the machine, which however is quite realistic and performed with state of the art tools, and on the computation of the "life-time performance" according to the procedure explained in Section 4.2.1. The sentence was slightly changed to make it clearer.

Again, these results are based on a model whose input are the 1xRev harmonics of in and out-of-plane blade root bending moment. The number of dofs in the model is irrelevant, as previously noted, and was eliminated to avoid confusion.

We have taken the opportunity to make several small editorial changes to the text, in order to improve readability. A revised version of the manuscript is attached to the present reply, with the main changes highlighted in red.

Best regards.
The authors

**Supplement:**

[revised manuscript text omitted]
}[\boldsymbol{\epsilon}_\theta] = \mathcal{E}[\boldsymbol{\epsilon}_\theta \boldsymbol{\epsilon}_\theta^T]$ (Cramer, 1946) writes

$$\mathrm{Cov}[\boldsymbol{\epsilon}_\theta] = \left(\boldsymbol{F}^T \boldsymbol{R}^{-1} \boldsymbol{F}\right)^{-1}. \tag{32}$$

This expression shows the interplay between noise $\boldsymbol{r}$ and sensitivity $\boldsymbol{F}$, captured by the term $\boldsymbol{R}^{-\frac{1}{2}}\boldsymbol{F}$: the higher the variance and/or the lower the sensitivity of the measurements with respect to the wind states, the worst the accuracy of the estimates.

The covariance $\mathrm{Cov}[\boldsymbol{\epsilon}_\theta]$ expressed by Eq. (32) is typically fully populated, as the errors of the estimates are correlated. To ease the understanding of the estimation problem, the SVD (Golub and van Loan, 1996) can be used to decouple the estimates. In fact, matrix $\boldsymbol{R}^{-\frac{1}{2}}\boldsymbol{F}$ can be factored as

$$\boldsymbol{R}^{-\frac{1}{2}}\boldsymbol{F} = \boldsymbol{U}\boldsymbol{\Sigma}\boldsymbol{V}^T, \tag{33}$$

where $\boldsymbol{U} \in \Re^{m \times m}$, $\boldsymbol{\Sigma} \in \Re^{m \times n}$ and $\boldsymbol{V} \in \Re^{n \times n}$, being $m$ the number of measurements and $n$ the number of wind state variables. Matrices $\boldsymbol{U}$ and $\boldsymbol{V}$ are orthonormal, i.e. $\boldsymbol{U}^T\boldsymbol{U} = \boldsymbol{U}\boldsymbol{U}^T = \boldsymbol{I}$ and $\boldsymbol{V}^T\boldsymbol{V} = \boldsymbol{V}\boldsymbol{V}^T = \boldsymbol{I}$, whereas $\boldsymbol{\Sigma} = \mathrm{diag}(\dots, 1/\sigma_i, \dots)$ is a diagonal matrix and $\sigma_i$ the standard deviation. Inserting Eq. (33) into Eq. (32), the covariance of the estimation error can be expressed as

$$\mathrm{Cov}\left[\boldsymbol{V}^T\boldsymbol{\epsilon}_\theta\right] = \boldsymbol{E}\left[\left(\boldsymbol{V}^T(\boldsymbol{\theta}_E - \boldsymbol{\theta}_R)\right)\left(\boldsymbol{V}^T(\boldsymbol{\theta}_E - \boldsymbol{\theta}_R)\right)^T\right] = \left(\boldsymbol{\Sigma}^T\boldsymbol{\Sigma}\right)^{-1} = \mathrm{diag}(\dots, \sigma_i^2, \dots). \tag{34}$$

This way, the problem is reformulated by the change of variables $\boldsymbol{\xi} = \boldsymbol{V}^T\boldsymbol{\theta}$, where $\boldsymbol{\xi}$ are statistically independent variables with diagonal covariance. This reformulation simplifies the interpretation of the structural observability of the problem. In fact, the $i$th column of matrix $\boldsymbol{V}$ linearly combines the wind parameters, mapping them into a new parameter $\xi_i$ with variance $\sigma_i^2$. Clearly, a high variance indicates a low level of identifiability of the associated linear combination of wind parameters.

This analysis also provides information on the dependence of loads on wind states. In fact, one can easily show that

$$\frac{\partial \boldsymbol{m}}{\partial \boldsymbol{\xi}} = \boldsymbol{R}^{\frac{1}{2}}\boldsymbol{U}\boldsymbol{\Sigma}. \tag{35}$$

Therefore, the analysis of $\boldsymbol{U}$ reveals on which linear combination of inflow parameters each load depends the most.

The same analysis can be applied to the nonlinear case, by linearizing Eq. (22) around a specific operating and wind condition and using $\boldsymbol{F} = \partial(\boldsymbol{F}_{\mathrm{NL}}\boldsymbol{\theta}_{\mathrm{NL}})/\partial\boldsymbol{\theta} = \boldsymbol{F}_{\mathrm{NL}}\partial\boldsymbol{\theta}_{\mathrm{NL}}/\partial\boldsymbol{\
[revised manuscript text omitted]

---

## Editor Decision (ED1)

**Journal:** WES
**MS No.:** wes-2017-23
**MS Type:** Research articles
**Submission Date** 2017-05-16
**Date Due** 2017-07-20
**Title:** "Wind inflow observation from load harmonics"
**Author(s):** Marta Bertelè et al.

Thank you for your answers. I'm satisfied of most of responses, I however still have a doubt on one point:

I clearly understand that your objective is to use strain gage outputs from field measurements to predict inflow states. The goal is very interesting and you have demonstrated its feasibility earlier on field measurements. No doubts on that. I'm just concerned on the present paper and on what dataset was used to demonstrate the present method. It is still not clear for me how many simulated "strain gage" inputs/outputs did you use to identify your model ? In other terms, where the artificial strain gages are located in your model and how many "cases" did you run for the results given p20? I think the authors can answer by giving the value of the numbers $N_{nodev}$, $N_{nodee}$ (Equation 16) and the number $N_{exp}$ (equation 36) used to obtain results p20, that are not given in the article ?

Also, by looking again at that figure 2, can you please add a colorbar ?

Best regards.

---

## Author Response (AR2)

**Reply to Reviewer (second review)**

**Reviewer** I'm just concerned on the present paper and on what dataset was used to demonstrate the present method. It is still not clear for me how many simulated "strain gage" inputs/outputs did you use to identify your model ? In other terms, where the artificial strain gages are located in your model?

**Authors** This was clarified by modifying the text is Sect. 2.3.5 as follows: "Loads were measured on the aeroelastic model in a blade-attached reference frame located at the root of each blade, thereby simulating the presence of strain gages measuring flapwise and edgewise bending moment components, which were then transformed into out and in-plane rotor components by using the blade pitch angle. Next, the out and in-plane loads were decomposed into their harmonics at the 1×Rev and 2×Rev by the Coleman transformation and used, together with the corresponding wind states, for identifying the linear and nonlinear models used throughout this work."

**Reviewer** and how many "cases" did you run for the results given p20? I think the authors can answer by giving the value of the numbers Nnodev, Nnodee (Equation 16) and the number Nexp (equation 36) used to obtain results p20, that are not given in the article ?

**Authors** This was clarified by modifying the text is Sect. 2.3.5 as follows: "Fully parameterized steady winds were generated at $N_{\mathrm{node}_V} = 10$ speeds $V = \{3, 4, 5, 6, 7, 8, 9, 11, 15, 19\}$ m/s, where for each different wind speed all possible 900 combinations of the following wind parameters were considered:

$$\phi = \{-16, -12, -8, -4, 0, 4, 8, 12, 16\} \text{ deg}, \tag{1a}$$
$$\kappa_v = \{0.0, 0.1, 0.2, 0.3, 0.4\}, \tag{1b}$$
$$\chi = \{0, 4, 8, 12\} \text{ deg}, \tag{1c}$$
$$\kappa_h = \{-0.1, -0.05, 0.0, 0.05, 0.1\}, \tag{1d}$$

resulting in $N_{\mathrm{exp}} = 9000$ available observations." Notice that these numbers apply to all results presented in the paper, as stated in the modified text ("...for identifying the linear and nonlinear models used throughout this work").

**Reviewer** Also, by looking again at that figure 2, can you please add a colorbar ?

**Authors** The color represents only the value of the function, i.e. the elevation of the plot, and hence a colorbar is not necessary. Although redundant, the use of the color helps understand the actual shape of the function. We modified the figure caption as follows to clarify this point: "The color indicates the value of the function, i.e. the elevation of the plot.".

Best regards
The Authors